# Enzobiotics—A Novel Therapy for the Elimination of Uremic Toxins in Patients with CKD (EETOX Study): A Multicenter Double-Blind Randomized Controlled Trial

**DOI:** 10.3390/nu14183804

**Published:** 2022-09-15

**Authors:** Anita Saxena, Sanjay Srinivasa, Ilangovan Veerappan, Chakko Jacob, Amol Mahaldar, Amit Gupta, Ananthasubramaniam Rajagopal

**Affiliations:** 1Department of Nephrology, Sanjay Gandhi Post Graduate Institute of Medical Sciences, Lucknow 226014, India; 2Center for Kidney and Diabetes, Department of Nephrology, Bangalore 560064, India; 3Department of Nephrology, KG Hospital, Coimbatore 641018, India; 4Department of Nephrology, Baptist Hospital, Bangalore 560024, India; 5Vision and Manipal Hospitals, Mapuca 403507, India; 6Apollo/Medics, Apollo Medics Kanpur Road, Lucknow 226005, India; 7Indian Statistical Institute, Coimbatore 641043, India

**Keywords:** *p*-cresol, indoxyl sulfate, eGFR, protein-bound uremic toxins (PBUTs), adversity ratio, SF-36-QoL

## Abstract

Design, participants, setting, and measurements: Predialysis adult participants with chronic kidney disease (CKD) and mean estimated glomerular filtration rate (eGFR) <45 mL/min per 1.73 m^2^) were recruited in 2019 to a multicentric double-blinded randomized controlled trial of enzobiotic therapy (synbiotics and proteolytic enzymes) conducted over 12 weeks. The primary objective was to evaluate the efficacy and safety of enzobiotics in reducing the generation of *p*-cresol sulfate (PCS) and indoxyl sulfate (IS), stabilizing renal function, and improving quality of life (QoL), while the secondary objective was to evaluate the feasibility of the diagnostic prediction of IS and PCS from CKD parameters. Results: Of the 85 patients randomized (age 48.76 years, mean eGFR 23.24 mL/min per 1.73 m^2^ in the placebo group; age 54.03 years, eGFR 28.93 mL/min per 1.73 m^2^ in the enzobiotic group), 50 completed the study. The absolute mean value of PCS increased by 12% from 19 µg/mL (Day 0) to 21 µg/mL (Day90) for the placebo group, whereas it decreased by 31% from 23 µg/mL (Day 0) to 16 µg/mL (Day 90) for the enzobiotic group. For IS, the enzobiotic group showed a decrease (6.7%) from 11,668 to 10,888 ng/mL, whereas the placebo group showed an increase (8.8%) from 11,462 to 12,466 ng/mL (Day 90). Each patient improvement ratio for Day 90/Day 0 analysis showed that enzobiotics reduced PCS by 23% (0.77, *p* = 0.01). IS levels remained unchanged. In the placebo group, PCS increased by 27% (1.27, *p* = 0.14) and IS increased by 20% (1.20, *p* = 0.14). The proportion of individuals beyond the risk threshold for PCS (>20 µg/mL) was 53% for the placebo group and 32% for the enzobiotic group. The corresponding levels for IS risk (threshold >20,000 ng/mL) were 35% and 24% for the placebo and enzobiotic groups, respectively. In the placebo group, eGFR decreased by 7% (Day 90) but remained stable (1.00) in the enzobiotic group. QoL as assessed by the adversity ratio decreased significantly *(p =* 0.00), highlighting an improvement in the enzobiotic group compared to the placebo group. The predictive equations were as follows: PCS (Day 0 = −5.97 + 0.0453 PC + 2.987 UA − 1.310 Creat; IS (Day 0) = 756 + 1143 Creat + 436.0 Creat^2^. Conclusion: Enzobiotics significantly reduced the PCS and IS, as well as improved the QoL.

## 1. Introduction

Despite advances in understanding the mechanisms responsible for causing renal disease and the advent of newer therapies for controlling modifiable risk factors, the decline in renal function is still inevitable. Traditional risk factors cannot completely explain renal outcomes in chronic kidney disease (CKD) patients. The intestinal microbiota has emerged as an important cause of the progression and complications of CKD [1]. Chronic kidney disease is a chronic uremic state, a perfect prescription for toxin formation and development of cardiovascular disease, contributing significantly to total mortality in dialysis patients. A dysbiotic gut microbiome has emerged as a prominent cause of the generation of uremic toxins which precipitate the progression of CVD [2].

Uremic toxins are biomarkers associated with decreased renal function. *p*-Cresol sulfate (PCS) and indoxyl sulfate (IS) contribute to CKD progression through renal tubular damage or tubulointerstitial fibrosis by activating free-radical production, upregulating nuclear factor (NF)-κB and plasminogen activator inhibitor type 1(PAI-1), and enhancing the expression of transforming growth factor beta 1 (TGFβ), tissue inhibitor of metalloproteinase (TIM), and pro-alpha 1 collagen [3].

In addition, IS is reported to be related to aortic calcification and vascular stiffness, associated with an increased risk of overall and cardiovascular (CV) mortality in patients with CKD through mechanisms of increasing oxidative stress in endothelial cells, the shedding of endothelial microparticles, impairing endothelial cell repair, and inducing vascular smooth muscle cell proliferation [4].

The first study to support the renal toxicity of PCS indicated that PCS was capable of resulting in renal tubular cell damage by inducing oxidative stress via activation of NADPH oxidase [5], with a similar mechanism exhibited by indoxyl sulfate (IS) [6]. In pre-end-stage renal disease (ESRD) patients, these results were further supported by Chi-Feng et al. [7], showing that the PCS level can predict CV events, as well as kidney function deterioration. These studies explicitly indicate that PCS is not only a vascular toxin but also a nephrotoxin.

Prolonged retention of undigested protein in the intestines triggers an immune response with increased inflammation in the gut and the formation of uremic toxins such as PCS and IS, which play an important role in the genesis of cardiovascular complications, progression of renal damage, and mortality in chronic kidney disease. Among all toxins known so far, IS and PCS have been reported to be involved in the development of CVD. 

PCS and IS are prototypic protein-bound uremic toxin molecules which are not only biomarkers for renal function, but have also been shown to contribute to the development of diseases [2]. The two toxins are very similar in their origin from the gut bacteria and their function as promoters of renal disease. Both are toxic products of protein metabolism [8], and they are bound to albumin at the Sudlow II site [9] with low dialytic clearance. PCS and IS play a significant role in renal metabolism [1,10], CVD, and mortality in renal patients [11,12].

The classical sources of uremic solutes such as dietary protein breakdown, as well as alternative sources such as the environment and herbal medicines, create uremic toxicity if not restricted. Many solutes with toxic capacity are produced in the intestine [13]; most research on uremic toxicity has focused on the retention and removal of these organic compounds [14].

During the natural progression of CKD, there is a shift in the microbiome composition and intestinal environment from a symbiotic to a dysbiotic state, caused by an increase in colonic protein fermentation, which results in an increase in microbiota-derived uremic toxins along with the diminution of carbohydrate fermentation, and the formation of shorty-chain fatty acids (SCFAs) is consequently compromised [15,16,17].

The amounts of nutrients entering the colon mainly depend on dietary intake and the efficiency of the assimilation process in the small intestine. Pancreatic inflammation and malabsorption are often associated with impaired protein digestion in the ESRD population [18]; consequently, changes in the composition of the intestinal flora or changes in intestinal production and absorption alter their serum concentration. Undigested proteins and dysbiosis of the gut in CKD result in the generation of IS from tryptophan and PCS from tyrosine [18,19]. These PBUTs contribute to worsening renal function and CKD progression [20]. PBUTs also increase CVD morbidity and mortality in CKD [20].

Earlier studies by Feng and Barretto [12], and Wu et al. [21] revealed the role of PBUTs (PCS and IS) as nephrotoxins and vasculotoxins, worsening aortic calcification and pulse wave velocity, thereby contributing to the progression of CKD, as well as CVD morbidity and mortality. Wu et al. established that PCS and IS may predict the risk of renal progression. The progression of CKD stages increases with the increase in toxins [21]. The dysbiotic microbiome, accompanied by a reduction in renal clearance, accumulation of undigested protein, and enhanced generation of toxins in the blood, results in a worsening of ESRD. Although this report provided a descriptive analysis of stages evidencing 10 mg/L PCS as a progressive case at an eGFR of 26 mL/min but 4 mg/mL PCS as a nonprogressive case at an eGFR of 48 mL/min, predictive equations for toxin levels were not provided.

### Why Would This Study Be Crucial?

The gut microbiome plays important roles in both the maintenance of health and the pathogenesis of disease. Gut microbiome dysbiosis results from an alteration of the composition and function of the gut microbiome, as well as a disruption of gut barrier function, which is commonly seen in patients with CKD. Therapeutic interventions should aim at restoring gut microbiome symbiosis. If proven effective, these interventions could have a significant impact on the management of CKD patients [20].

Undigested proteins from small intestine get fermented in dysbiotic colon triggering immune response inducing inflammation. PCS from Tyrosine, and IS from Tryptophan induces inflammation contributing to CKD complications (Figure 1). Earlier studies targeted the dysbiotic gut to improve uremic toxins in CKD. However, the targeting of impaired protein assimilation (digestion and absorption) in CKD [22] and the movement of unmetabolized protein to the colon leading to the generation of PBUTs [23] have not been sufficiently addressed. Studies on synbiotics (probiotics + prebiotics) resulted in significant improvements in the serum urea, creatinine, hsCRP, TNF-alpha, and quality of life of dialysis patients [24]. Another study established the role of proteolytic enzymes in metabolizing proteins and improving protein assimilation, which was reflected as higher serum albumin levels in peritoneal dialysis patients [25]. A preclinical trial established the superiority of enzobiotics (synbiotics + proteolytic enzymes) over synbiotics alone and proteolytic enzymes alone in gentamycin-induced CKD in Wister rats [26]. Hence, the rationale is that the simultaneous application of synbiotics and proteolytic enzymes (enzobiotics) is synergistic [26]. Since inflammation and oxidative stress are evident in moderate stages of CKD, the key hypothesis is that controlling toxin levels can reduce CKD complications and slow CKD progression. Enzobiotics can modulate the intestinal microbiota and improve the absorption of proteins in the small intestine to potentially prevent the formation of protein-bound uremic toxins (PBUTs) from the intestinal microbial metabolism of aromatic amino acids, thus benefiting the quality of life of patients with CKD. Therefore, the primary objective of this EETOX study was to evaluate the efficacy and safety of enzobiotics in reducing the generation of *p*-cresol sulfate and indoxyl sulfate, stabilizing renal function, and improving quality of life (QoL), while the secondary objective was to evaluate the feasibility of the diagnostic prediction of IS and PCS from CKD parameters.

## 2. Materials and Methods

This was a double-blinded, placebo-controlled multicentric study. The minimum sample size of subjects was determined as *n* = 20 in each group to detect an accuracy of ±6–9 µg/mL for PCS and of ±7–10 ng/mL for IS with a 95% confidence level and an 80% power of correct detection. Considering possible attrition, a total of 85 subjects were recruited from Coimbatore (23 from Tamilnadu), Bangalore (25 from Karnataka), Lucknow (30 from Uttarpradesh), and Goa (seven from Goa). Computer-generated random numbers were used to allocate subjects to both groups. Group A was supplemented with enzobiotics (*n* = 50), while Group B was administered a placebo (*n* = 35). The distribution of subjects was based on eGFR.

### 2.1. Study Site, Design, and Participants

Patients were recruited from March 2019 to August 2019. The number of patients recruited according to stage was as follows: stage 5 (*n* = 46), stage 4 (*n* = 21), stage 3 (*n* = 13); Day 90 dropouts were excluded (*n* = 5). 

### 2.2. Ethical Consideration

The trial was approved by the ethics committees of the participating institutes and was registered with the central drugs standard control organization of the Clinical Trial Research Institute of the Government of India. The study was conducted in accordance with the Declaration of Helsinki and approved by the Ethics Committee of the Sanjay Gandhi Postgraduate Institute of Medical Sciences, Lucknow, KG Hospital, Coimbatore, Tamil Nadu, Baptist Hospital, Bangalore, Karnataka, and Vision and Manipal Hospitals, Goa (protocol code Nizy-Byt-Mb-18). All study-related documents were reviewed by the institutional ethics committees, and eligible subjects were enrolled into the study only after obtaining written consent. The study was prospectively registered in CTRI (CTRI/2019/01/017070) with protocol code Nizy-Byt-Mb-18.

### 2.3. Study Design

Patients were screened (Visit 0) for initial eligibility, followed by additional screening at Visit 1 (Day 0), Visit 2 (Day 45), and Visit 3 (Day 90). For each visit, the window was seven days; subjects reporting beyond 97 days were not considered for final analysis. Drug logs were recorded as per the Declaration of Helsinki. Concomitant medications were not changed. The investigation product (IP) was dispensed at each visit, and subjects were asked to return used and unused medication to the study facility at each visit. The subjects followed a CKD diet prescribed by a renal dietician (protein 0.6 g/kg/day; energy 30–35 kcal/day; sodium 2.0 g/day; potassium <1 mEq/kg/day; phosphorus 800–1000 mg/day; calcium 1500–2000 mg/day), with fluid restrictions according to renal function. The diet pattern of the subjects in both enzobiotic and placebo groups was not changed after recruitment in the study to prevent any introduction of bias and to determine the effect of enzobiotic therapy with a CKD diet.

A schema of the study is given in Figure 2 following the CONSORT guidelines.

#### 2.3.1. Inclusion Criteria

The inclusion criteria were as follows: subjects aged 18–70 years in CKD stages 3, 4, and 5 that were willing to come for regular follow-ups and that provided written informed consent.

#### 2.3.2. Exclusion Criteria

The exclusion criteria were as follows: subjects below 18 years of age, pregnant and lactating women, patients with autoimmune disease or active infection over the past month, subjects with hepatic impairment (SGOT or SGPT levels >3 times the upper limit), with unstable cardiovascular history, already on pre/probiotics or proteolytic enzymes, and with severe systemic illness. No exclusions were applied according to BMI.

All adverse (AEs) and serious adverse events (SAEs) were recorded.

#### 2.3.3. Intervention

The treatment group was administered interventional proprietary enzobiotic capsules containing probiotics (*Lactobacillusacidophillus* 100 mg, *Bifidobacteriumlongumm* 100 mg, *Streptococcus thermophilus* 50 mg), prebiotics (FOS 100 mg), and proteolytic enzymes (150 mg with an activity 355,000 IU) in 500 mg capsules. The placebo group was administered maltodextrin capsules (500 mg).

Groups received the enzobiotic or placebo capsules 5 min before each major meal. Each subject was allotted a unique identification number.

#### 2.3.4. Blinding Procedure

The double-blinding conditions were applied by keeping both patients and principal investigators blinded to the treatment (placebo or enzobiotics) in order to avoid bias. The two capsules were identical in terms of color, shape, size, strips, and boxes. The groups were coded centrally using computer-generated random numbers as per a randomization schedule.

### 2.4. Analysis Methods for PCS, IS, and Biochemical Parameters

The STARD guidelines were followed for the diagnostic accuracy of PCS and IS (see Appendix A), according to an NABL-accredited Laboratory, Notrox Research Private Limited Bengaluru.

#### 2.4.1. Diagnostic Variables Studied on Day 0 and Day 90

Thirty-four diagnostic variables were measured: renal function parameters, i.e., serum creatinine, blood urea nitrogen (BUN), urea, sodium (Na), serum potassium (K), serum phosphates (P), uric acid (UA), and eGFR (A—MDRD and W—Cockraft–Gault); cardiac parameters, i.e., systolic blood pressure, diastolic blood pressure, heart rate, pulse rate (PR), respiratory rate, total cholesterol (TC), high-density cholesterol (HDL), low-density cholesterol (LDL), and triglycerides (TGs); hematological parameters, i.e., hemoglobin, platelet count, hematocrit, white blood corpuscles, red blood corpuscles, and serum albumin. Hypertension was defined as systolic blood pressure >130–139 mm Hg and diastolic blood pressure >80–89 mm Hg, while diabetes mellitus was defined as HbA1C >6.5%. High-sensitivity C-reactive protein (hsCRP) and hepatic functions (serum glutamic–oxaloacetic transaminase (SGOT), serum glutamic–pyruvic transaminase (SGPT), and TB (total bilirubin)), along with nutritional status parameters such as height, body weight, and body mass index (BMI), were recorded. Anthropometric measures were taken as per standard techniques. Biochemical parameters collected from all centers were tested in a centralized NABL-accredited laboratory (SRL Ranbaxy Laboratory, Mumbai, India) to avoid bias. Reference ranges (cutoffs) for biochemical parameters were established according to the same laboratory.

For patient evaluation, the medical history was taken face-to-face in the outpatient department. Physical examination was performed by the consultant. Vitals were taken by a nurse. Information was captured on case report forms. Patients were recruited without gender bias on the basis of their willingness to take the trial drug or placebo.

#### 2.4.2. Measurement of Quality of Life

To determine the effect of enzobiotics on quality of life, the SF-36 questionnaire KDoQI was used. Adversity was defined according to the proportion of relatively bad scores. These 36 questions described the health status of patients in both groups according to the self-assessed quality of living on Day 0 and Day 90. The items are categorized into seven components: general wellbeing, daily activity limitations, problems in the last four weeks, emotional problems in the last four weeks, feelings in the last four weeks, and health. The analysis was verified for internal consistency using Cronbach’s alpha before comparing the adversity ratio between groups [27]. The standard deviation was also compared for consistency between groups. Analysis was conducted using the binomial proportion of the p-chart for each question, and the reduction in item scores due to enzobiotics was evaluated.

### 2.5. Statistical Analysis

#### 2.5.1. Descriptive Analysis

The parametric *t-*test for the differences of means was used since the Day 90/Day 0 ratio followed a normal distribution. The absolute values of PCS followed normality, while those of IS followed a log-normal distribution. The proportion of patients beyond the risk cutoff level were estimated using capability analysis. The nonparametric Mann–Whitney test was used in non-normal cases. The improvement ratio in PCS and IS on Day 90 compared to Day 0 was recorded for the enzobiotic and placebo groups (1, no change; >1, adverse outcome; <1, improvement). The normality and homogeneity of variance (F-ratio) for initial observations (Day 0) in both groups were analyzed for randomness. The cutoffs for risk of CKD progression and CVD in the study were established at PCS > 20 µg/mL and IS > 20,000 ng/mL. These values were determined from the multiple regression for the prediction of toxin on Day 90. The improvements in PCS and IS were compared with the normal range using capability analysis after treatment to identify potential risks in both groups. A two-sample *t*-test was used to test the significance of differences between both groups in terms of the Day 90/Day 0 ratio, after verifying for normality.

The subjects not meeting the criteria for analysis (see Table 1) were discarded to maintain the veracity and validity of observations for reliable statistical comparison.

#### 2.5.2. Predictive Analysis

The dimensional reduction of 34 interdependent diagnostic variables was performed to predict the toxin using principal component analysis, eigenvalues (scree plot), eigenvectors (loading plot), and multiple regression analysis.

Furthermore, the predictors were externally validated on 585 patients for the correlation of creatinine with platelet count and uric acid in PCS. Similarly, IS was predicted on the basis of creatinine, phosphorus, and urea.

We used the following cutoffs for risk of CKD progression: PCS > 20µg/mL and IS > 20,000 ng/mL on Day 90.

## 3. Results

The study recruitment and attrition statistics according to the Strengthening the Reporting of Observational Studies in Epidemiology (STROBE) guidelines are shown in Table 1; of the 85 subjects recruited, 50 completed the study.

Table 2 provides the baseline (Day 0) demographic data and clinical data, which were comparable across both groups, with no significant differences in terms of age, gender, BMI, blood pressure, blood urea, plasma creatinine, complete blood count, electrolytes, phosphorus, hsCRP, uric acid, and lipid profile, confirming the random nature of the allocation. Both groups had comorbid conditions (placebo group: both HT and DM *n* = 8, neither DM nor HT *n* = 2, DM *n* = 2, HT *n* = 9; enzobiotic group: both HT and DM *n* = 14, neither DM nor HT *n* = 5, DM *n* = 10, HT *n* = 0). Hypertension was defined as systolic blood pressure >139 mmHg, while diabetes mellitus was defined as HbA1C > 6.5%. Both systolic blood pressure (*p* = 0.084) and diastolic blood pressure (*p* < 0.05) were better controlled in the enzobiotic group than in the placebo group on Day 90.

The outliers were identified as patients with PCS values 1.5 times beyond the upper (Q3 + 1.5 × IQR) or lower (Q1 − 1.5 × IQR) interquartile range. The interquartile ranges are presented in Table 3 and compared in Table 4.

### 3.1. Risk of CKD Progression

The risk of CKD progression in this study was defined as >20 µg/mL for PCS and >20,000 ng/mL for IS, as determined from multiple regression for the prediction of toxin on Day 90. The PCS distributions (Appendix A) and IS distributions (Appendix A: Appendix A) on Day 0 and Day 90 highlight the random nature of both groups. The proportions of subjects beyond the threshold levels are given in Figure 3.

The risk of progression to a higher CKD stage according to the PCS threshold was 53% for the placebo group but 32% for the enzobiotic group, while that according to the IS threshold was 35% for the placebo group but 24% for the enzobiotic group.

### 3.2. Quality of Life

The health-related quality of life was examined using the SF-36 questionnaire as per the SQUIRE guidelines. Both groups presented a low score at baseline, which improved on Day 90. The proportion of subjects with relatively bad quality of life scores (adversity ratio) is given in Table 5.

### 3.3. Prediction of Toxins Using Multiple Regression Method

The vital variables are shown as a matrix plot in Figure 4 describing their relationship with toxins. The matrix plot depicts the association of uremic toxins PCS and IS with routine uremic CKD parameters. The interdependency of PCS with routine uremic diagnostic variables including platelet count, uric acid, and creatinine is shown. The statistical significance of uremic parameters PCS is established in Table 6, and the significance of the biomarkers as predictors is presented in Table 8. Similarly, the interdependency of IS with creatinine, urea, and phosphorus is shown, while the statistical significance is established in Table 6 and the significance of the biomarkers as predictors is presented in Table 10. The matrix plot depicts how the change in PBUTs was influenced by the changes in the levels of routine parameters. The absolute comparison of simple averages could not highlight the differences in PBUTs within 90 days; however, they were influenced by routine testing biomarkers, thus enabling their prediction. This allowed establishing threshold levels for PBUTs to compare the risks of placebo compared to enzobiotics.

The association of PBUTs with each routine parameter was examined, and the significance of the association is given in Table 6 for PCS and IS.

The dimensional reduction of the 34 study variables for diagnostic prediction was carried out using principal component analysis, as shown in Table 7.

The relationships of PCS and IS with 11 predictor variables are shown as a matrix plot in Figure 4, where the red curve depicts the relationship pattern. A total of 82% of the structural variation was explained by 11 principal variables; nine negative contrast variables (creatinine, BUN, urea, phosphorus, HDL, uric acid, heart rate, pulse rate, and platelet count) and two positive contrast variables (GFR and RBC) were identified as significant predictors for PCS on Day 0 from the multiple linear regression (Table 8).

### 3.4. Prediction Equations

The results show that PCS was reduced by 0.05 µg/mL and 3 µg/mL for every unit increase in platelet count and uric acid, respectively. Enzobiotics reduced the PCS from 28.62 to 22.92 (20%).
PCS (Day 0) = −5.97 + 0.0453 PC + 2.987 UA − 1.310 Creat
Placebo PCS (Day 90) = 28.62 − 0.0542 PC + 0.283 PCS (Day 0)
Enzobiotic PCS (Day 90) = 22.92 − 0.0542 PC + 0.283 PCS (Day 0)

The diagnostic prediction (Appendix A) revealed that when PC decreased below 250, PCS increased beyond 20 µg/mL on Day 0. In the enzobiotic group, on Day 90, when the platelet count was between 200 and 400, the PCS decreased to below 15 µg/mL; in the placebo group, when the platelet count was below 200, *p*-cresol increased beyond 20 µg/mL.

The external validation (*n* = 585; see Appendix A
Appendix A) establishing the relationship among creatinine, uric acid, and platelet count is given in Table 9.

It was found that the variables uric acid and platelet count had a significant effect on creatinine.

IS relationship at Day 0: The association of IS with significant uremic parameters according to the diagnostic prediction is shown in Appendix A.

It was found that 50% of the variation in IS could be explained by creatinine on Day 0. The prediction equation is as follows:IS (Day 0) = 756 + 1143 Creat + 436.0 Creat^2^

The multiple linear regression results for the prediction of IS on Day 90 are given in Table 10.

The regression equations on Day 90 for the placebo and Enzobiotic groups are as follows:Placebo IS (Day 90) = −16,270 + 4689 Creat (Day 0) − 242.71 Urea (Day 0) + 3946 P (Day 0) + 1.206 IS (Day 0)
Enzobiotics IS (Day 90) = −19,551 + 4689 Creat (Day 0) − 242.71 Urea (Day 0) + 3946 P (Day 0) + 1.206 IS (Day 0).

A total of 72.58% of the variation in IS after 90 days could be explained by the above uremic variables (i.e., serum creatinine, urea, and phosphorus).

According to the diagnostic prediction, when the serum creatinine, serum urea, and serum phosphorus were increased by 1 mg/dL, the IS increased by 4689 ng/dL, decreased by 243 ng/dL, and increased by 3946 ng/dL, respectively.

## 4. Discussion

The present study was conducted to examine the effect of Enzobiotic supplementation on uremic toxins PCS and IS in pre-dialysis CKD patients. PCS and IS are documented independent markers of renal dysfunction, CKD progression, cardiovascular morbidity, and all-cause mortality [28,29,30].

Taking into account the poor-quality, inconsistent, and nonsignificant results of previous trials, this EETOX study represents the first successful attempt of its kind, where protein assimilation was increased using proteolytic enzymes in combination with synbiotics (i.e., Enzobiotics) for the restoration of a normal eubiotic state in the intestine of CKD patients.

This study enrolled subjects in a homogeneous manner for the Enzobiotic and placebo groups (*p* = 0.207). PCS showed a significant reduction in the Enzobiotic group. The Day 90/Day 0 ratio for PCS was reduced by 23% with Enzobiotics, while it increased by 27% in the placebo group. A similar albeit less impressive trend was seen for IS levels; the absolute median value was reduced by 6.7% in the Enzobiotic group compared to an 8.8% increase in the placebo group. The Day 90/Day 0 ratio for IS in the placebo group was increased by 20% compared to a 0.3% increase in the Enzobiotic group, highlighting the striking benefits of treatment.

Our study shows that Enzobiotic therapy could reduce and/or maintain the concentrations of IS and PCS. For predicting PCS and IS levels in the serum, all 34 variables were found to be interdependent (Appendix A). A total of 82% of the structural variance among variables was explained by 11 principal components, nine of which exhibited negative contrast (creatinine, BUN, urea, phosphorus, HDL, uric acid, heart rate, pulse rate, platelet count, eGFR, and RBC) and two of which exhibited positive contrast (heart rate and platelet count), as shown in Appendix A. The concentration of uremic toxins was predicted on the basis of these variables using regression analysis. When PC increased by one unit and UA increased by three units while maintaining creatinine at 1.3 units, *p*-cresol increased by 0.05 units. This relationship can be used to predict the initial *p*-cresol. When dealing with any stage of CKD or any other health condition, the gut is of paramount importance. Furthermore, abnormalities in platelet function are central to the development of both thrombotic and hemorrhagic complications [31], and CKD patients are at highly increased risk of CVD complications. 

A hallmark of CKD is the accumulation of uremic retention solutes in the blood due to decreased kidney function, which poses a potential threat to the cell physiology of blood cells and platelets. Platelet count as a biomarker predicted a 20% reduction in initial PCS level on Day 90 of Enzobiotic therapy. PCS was reduced by 29 µg/mL in the placebo group and by 23 µg/mL in the Enzobiotic group.

Using multiple linear regression, this study determined the upper critical limit (UCL) as 20 µg/mL for PCS and 20,000 ng/mL for IS. The Enzobiotic group had all patients within the UCL on Day 90 irrespective of baseline values unlike the placebo group. The threshold levels of PCS and IS as established by Constance and Baaten in 2021 support these findings [32].

The relationship of PCS with platelet count (*p* = 0.018), uric acid (*p* = 0.001), and creatinine (*p =* 0.047) could be used to predict the PCS value. When the platelet count was below 2.5 Lac/mm^3^, the PCS was beyond 20 µg/mL on Day 0. After 90 days of treatment, all subjects in the Enzobiotic group had a platelet count above 2.5 Lac/mm^3^ and PCS below 20 µg/mL.

The validation of platelet count as a predictor variable using external data (*n* = 585; see Appendix A) demonstrated that, although creatinine level increased as a function of age, gender, and CKD stage, it decreased by 0.3 mg/dL for every million unit increase in platelet count. It was found that both uric acid and platelet count had a significant effect on creatinine (*p* < 0.001 and *p* < 0.018, respectively). The EETOX study clearly demonstrates the direct effects of uremic toxins on platelet count. Previously published studies have also shown the inverse relationship of uremic toxins with platelet count [32]. CKD was also shown to lead to an increased risk of thrombotic and cardiovascular complications [33,34].

A predictive equation for PBUTs based on their association with routine kidney laboratory parameters is necessary due to the cost and non-availability of testing in underdeveloped countries. The application of synbiotics (probiotics + prebiotics) with proteolytic enzymes evidenced the strong relationship of PBUTs with routine kidney lab parameters (uric acid, creatine, and platelet count for PCS; creatinine, urea, and phosphorus for IS). PBUTs are highly specific markers for CKD. They determine the severity of the disease, as well as enable a prognosis.

At 90 days, 72.6% of the variation in IS could be explained by the creatinine, urea, and phosphorus levels. No other study has recorded such an association. The RBC count showed a negative linear correlation with IS (*p =* 0.022), which is in line with previous studies [34,35,36]. However, with Enzobiotic treatment, the changes in IS levels were greater. When IS exceeded 20,000 ng/mL, the RBC decreased below 4. These results would translate to better erythropoiesis with Enzobiotics in CKD; however, this statement requires validation in future studies.

Enzobiotic use did not change the urea and creatinine levels over 90 days, suggesting a stabilization of CKD and a potential to delay in progression. The creatinine level decreased by 0.3 mg/dL for every million unit increase in platelet count. In the absence of facilities to test for uremic toxins, IS levels can be predicted as a function of creatinine (IS increase of 1143 ng/mL for every 1 mg/dL increase in creatin). 

While the serum urea level was around 90 mg/dL in both groups, a change in urea changed IS levels; when the urea level was below 120 mg/dL, IS was below 20,000 ng/mL. Three variables (creatinine, PC, and RBC) had high predictability for the potential risk of high levels of uremic toxins. Previous studies support the predictive value of PCS and IS in patients with CKD [30]. Interpretation of the results may have been affected by the different cutoff values of PCS and IS, as well as low albumin concentrations. A systematic review of the literature after excluding studies biased by albumin binding considerations revealed that PCS and IS indeed play a role in vascular and renal disease progression [37].

The exact mechanisms via which elevated levels of IS and PCS contribute to CVD and mortality, however, have not been elucidated. Studies suggest that IS and PCS may suppress the activity of activated leucocytes, inhibit the release of platelet-activating factor by macrophages, and contribute to endothelial dysfunction and oxidative stress [38,39,40].

The odds ratio on Day 90 showed that hypertension worsened (100%) in the placebo group (OR; CI 0.5 to 8.5) compared to Enzobiotics. Therefore, the administration of Enzobiotics may reduce serum toxins, subsequently reducing the burden of cardiac function. 

Earlier studies [41,42] also indicated a better response for PCS than for IS. This clinical trial was performed with the expectation of reducing the toxin levels and improving QoL in CKD patients. This study used the SF-36 (KDoQI) scale to evaluate QoL, with Cronbach’s alpha values [27] confirming the internal consistency (0.9237 for placebo and 0.9364 for Enzobiotics) and reliability. To ensure the robusticity of the study, the patients were advised to adhere to the CKD diet plan to ensure that the gut microbiome was not altered due to the introduction of new food groups to the diet [43]. To determine the effect of Enzobiotic supplementation on quality of life in the present study, SF-36 (KDoQI) was used to assess health concepts relevant across age, disease, and treatment groups in adults. The SF-36 domains, particularly the physical functioning scores, showed good evidence of the impact of the intervention on patients with CKD. Supplementation with Enzobiotic therapy for 90 days led to an overall improvement in all 11 questions related to general wellbeing, along with improvements in terms of emotional problems, feelings, and health. The SF-36 (KDoQI) questionnaire has been validated on Indian CKD patients, with results showing that the tool is sound and has good internal consistency, convergent validity, and discriminant validity [44].

The QoL consistently and significantly improved in the Enzobiotic group, which could translate to a delay in the initiation of dialysis. However, this inference needs validation on a larger pool of patients.

We believe that several findings support probiotic supplementation in CKD. CKD-related alterations of the gut microbiota are related to dietary restrictions to prevent hyperkalemia [45]. In the dialysis population, excessive ultrafiltration volumes and/or intradialytic hypotension causing transient intestinal ischemia can aggravate the dysfunction and permeability of the gut barrier. A permeable gut barrier promotes the translocation of bacteria and endotoxins through the intestinal wall, thereby activating innate immunity and triggering a local inflammatory process, which contributes to perpetuating damage to the gut barrier and increasing oxidative stress [38]. The frequent use of antibiotics can potentially modify the bacterial community, adding to imbalance in the gut flora. Studies have shown that probiotic treatment can lead to 22–28% reductions in PCS in the analysis of antibiotic-free patients [46,47,48]. Hence, due credence should be given to the results of probiotic supplementation if antibiotic therapy is being given to a patient. Although no significant changes in BUN and hemoglobin were found in a metanalysis of RCTs after treatment with probiotics, there is evidence that probiotic supplementation can reduce PCS and increase IL-6 in patients with CKD.

Additionally, some prescriptions for antibiotics can alter the intestinal microflora, while others, such as phosphorus binders and ion exchange resins, may also slow intestinal transit time. PCS and IS are two among 80 known toxins; studies have shown that the serum concentrations of IS and PCS in CKD patients are 42 and 17 times higher, respectively, than in healthy individuals. Furthermore, because they are bound to albumin, they are only approximately 30% eliminated by hemodialysis [49].

Patients with CKD are also at risk of developing encephalopathy and cognitive impairment. A positive correlation between fibroblast growth factor 23 (FGF-23) and IS serum levels, which may imply a link between this molecule and metabolic bone disease in uremic patients, has been reported. Furthermore, a relationship between IS and CKD-related anemia has been observed since it diminishes erythropoiesis, hampers the activity of erythropoietin, and enhances the programmed cell death of red blood cells (eryptosis). Increased inflammatory biomarkers in stage 3–4 CKD patients, such as glutathione peroxidase and interleukin-6, have been reported. Vascular inflammation is involved in the pathogenesis of thrombotic complications. Controlling inflammation, dysbiosis, and oxidative stress can prevent the occurrence of CVD in CKD. IL-6 acts both as a pro- and an anti-inflammatory cytokine depending upon whether it activates the signal transducer and activator of transcription (STAT1) [50].

Dysbiosis and the formation of uremic toxins are strongly implicated as the cause of cardiovascular disease (CVD) in CKD. Increased levels of IS are associated with enhanced oxidative stress in endothelial cells, vascular smooth muscle cell proliferation, vascular stiffness, peripheral vascular disease, aortic calcification, and both overall and cardiovascular mortality in patients with CKD [51]. IS exerts a cardiac profibrotic effect, favoring myocardiocyte hypertrophy and predisposing to atrial fibrillation. Hence, it is presumed that the intervention with Enzobiotics might reduce the CVD burden by recreating balance in the gut microflora.

Direct effects of uremic toxins on platelet function have been described. CKD patients are at increased risk of thromboembolic complications, including myocardial infarction, stroke, deep-vein thrombosis, and pulmonary embolism, which affect clinical outcome and survival. This prothrombotic phenotype is ascribed to CKD-associated dysbiosis (indolic compounds induce platelet hyperactivity), giving rise to the phenomenon known as the thrombolome (uremic toxins that enhance thrombosis by increasing tissue factor expression, platelet hyperactivity, microparticles release, and endothelial dysfunction), which is spontaneously induced. In the presence of a low concentration of collagen and thrombin, platelet adhesion and aggregation are enhanced by P-selectin and GPIIb/IIIa expression, as well as a key regulator of platelet activation—oxidative stress. Platelet activity is enhanced by the IS-induced production of ROS-mediated p38 mitogen-activated protein kinase (p38 MAPK) [52].

Prebiotics are the food of the microbiota, which can be obtained fresh fruits, vegetables, and whole grains. In general, a diet that contains more fresh foods and fiber with reduced protein intake can provide optimal results. Since CKD patients have restrictions with respect to the consumption of fruits and vegetables, the development of hyperkalemia, and the prevention of hyperphosphatemia, a combination of proteolytic enzymes combined with pre- and probiotics can plausibly prevent dysbiosis and maintain gut health, and thus prevent CKD progression. Studies have shown that the intake of prebiotic inulin, enriched with oligofructose, could significantly reduce the serum concentrations of PCS and IXS [53].

In advanced renal failure, the removal of ureic toxins is made possible by different types of extracorporeal treatment; however, eliminating uremic toxins in the early stages of CKD is challenging. The intestinal production of toxins can be reduced by influencing dietary habits, improving gastroparesis, or orally consuming absorbents. The colonic transit time is a target in CKD which cannot be mediated by pharmacological therapies; however, prebiotics can reduce colonic transit time, in addition to other effects. The use of pre/probiotics should be started in the early stages of kidney failure and maintained for periods longer than four months [54].

### 4.1. Strength of the Study

This is the largest randomized double-blind trial of synbiotics in CKD described so far, with 50 subjects completing the study. The protocols were strict and strongly adhered to. Statistical analysis was extensive and unique. For the first time, a more rational approach to combining the synergistic benefits of proteolytic enzymes with synbiotics (Enzobiotics) was attempted with encouraging outcomes. The reliability of the data is evident, as the baseline parameters of the subjects were comparable in both groups. In the absence of available testing facilities for IS and PCS, this study found associations with routine clinical parameters and developed novel prediction equations.

### 4.2. Limitations of the Study

Only 50 patients were analyzed in the study and the duration of the study was short, considering the gradually progressive nature of CKD. In the future, studies using a randomized block design are recommended, with each block representing cohorts of patients (pre-existing DM, HT, both DM and HT, and neither DM not HT) observed six months before and after intervention.

## 5. Conclusions

Our study shows that Enzobiotic therapy can potentially maintain renal function (GFR), and increase red blood cell and platelet count by controlling the generation of IS and PCS. Our EETOX study is the first of its kind to successfully apply proteolytic enzymes in combination with synbiotics for the restoration of a eubiotic state of the gut in CKD.

## Figures and Tables

**Figure 1 nutrients-14-03804-f001:**
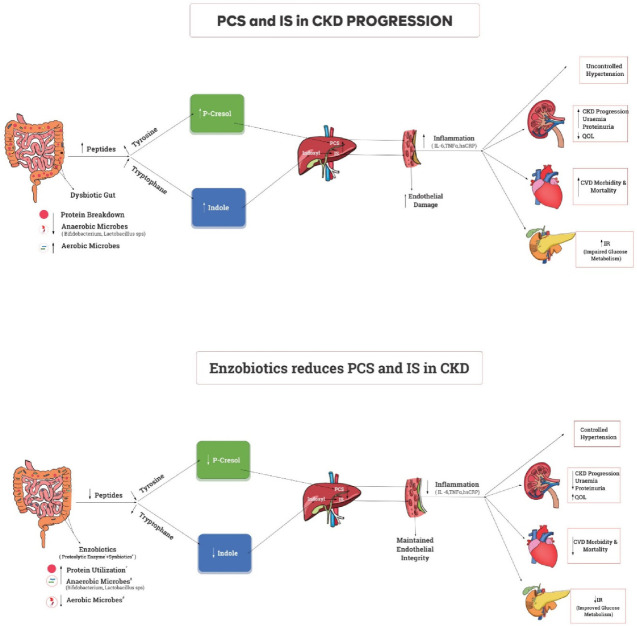
A dysbiotic gut with inadequate protein metabolism contributes to the increased inflammatory state responsible for CKD progression, morbidity, and mortality. Supplementation of enzobiotics (synbiotics + proteolytic enzymes) with food ensures complete protein metabolism in a eubiotic environment, thus reducing the inflammatory state and leading to better outcomes in CKD rat model [27].

**Figure 2 nutrients-14-03804-f002:**
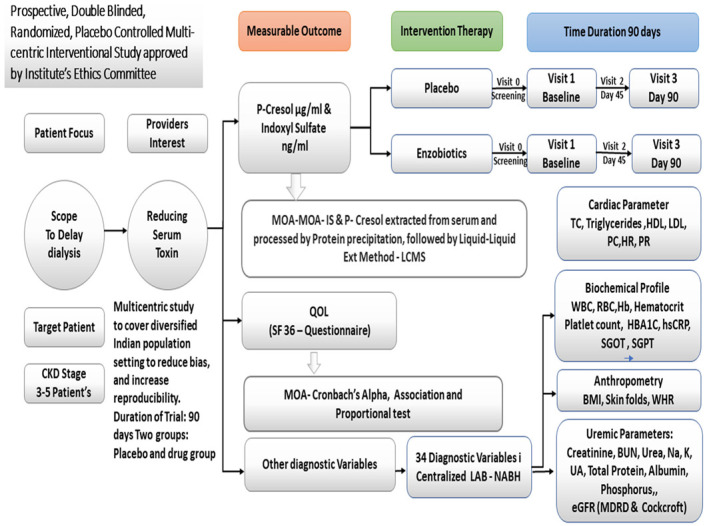
Schema of the Study. MOA, method of analysis; QoL, quality of life; TC, total count; HDL, high-density lipids; LDL, low-density lipids; SGOT/SGPT, serum glutamic–oxaloacetic transaminase; SGPT, serum glutamic–pyruvic transaminase; BUN, blood urea nitrogen; Na, sodium; K, potassium; P, phosphorus; UA, uric acid; PC, platelet count; HR, heart rate; PR, pulse rate; NABH, National Accreditation Board for Hospitals and Healthcare Providers. Diversified Indian population: 85 subjects were recruited from Coimbatore (23 Tamilnadu), Bangalore (25 from Karnataka), Lucknow (30 from Uttarpradesh), and Goa (seven from Goa).

**Figure 3 nutrients-14-03804-f003:**
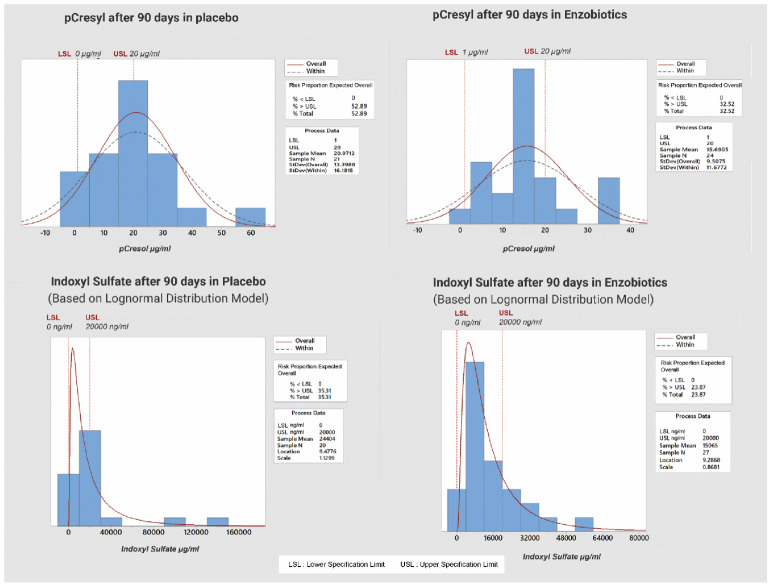
Risk of aggravation of CKD beyond threshold value of PCS and IS (for placebo and enzobiotic groups) on Day 90.

**Figure 4 nutrients-14-03804-f004:**
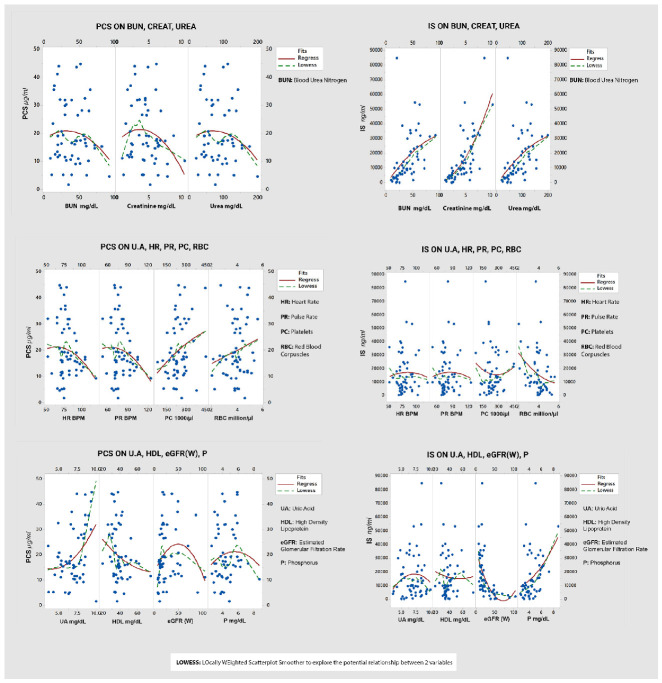
Matrix plots showing the relationship of PCS and IS with uremic parameters. BUN, blood urea nitrogen; CREAT, creatinine; UREA, urea; HR, heart rate; PR, pulse rate; PC, platelet count; RBC, red blood cells; UA, uric acid; HDL, high-density lipoproteins; eGFR(W), estimated glomerular filtration rate (Cockcroft–Gault method).

**Table 1 nutrients-14-03804-t001:** Study recruitment and attrition statistics.

	Enzobiotic Group	Placebo Group
Total patients recruited at start of study	50	35
Lost to follow-up	3	1
Consent withdrawn	1	6
Logistic delays (sample reaching central lab after >48 h)	4	2
Protocol violation (missed follow-up visit)	5	3
Follow-up after predefined acceptable study window for final visit, i.e., reported after 97 days.	8	2
Total dropouts	21	14
Total patients who completed study	29	21

**Table 2 nutrients-14-03804-t002:** Baseline demographics and lab data.

Variables		Placebo Group		Enzobiotic Group		Significance
	Statistics	Day 0 (*n* = 21)	Day 90 (*n* = 21)	Day 0 (*n* = 29)	Day 90 (*n* = 29)	*p*-Value
		Mean ± SD Median CI	Mean ± SD Median CI	Mean ± SD Median CI	Mean ± SD Median CI	Placebo vs. Enzobiotics on Day 90
Age	Mean CI for mean	49 ± 15	49 ± 15	54 ± 10	54 ± 10	0.176
		42–56	42–56	50–58	50–58	
Body mass index	Mean	26.08 ± 6.921	26.28 ± 7.04	25.61 ± 4.52	25.57 ± 4.55	0.689
	CI for mean	22.93–29.23	23.073–29.480	23.89–27.33	23.838–27.300	
Blood pressure—systolic *	Median	130 ± 22.6	138 ± 18.89	140 ± 15.29	130 ± 26.54	0.084
	CI for median	128.37–138.65	130–144.33	130–144	128.34–140.0	
Blood pressure—diastolic	Mean	84.05 ± 15.53	85.19 ± 12.89	90 ± 11.77 *	75 ± 9.982 *	0.034 *
	CI for mean	76.98–91.12	79.32–106	78.84–90	70–80	
Blood urea	Mean	89.63 ± 46.96	92.9 ± 48.9	81.33 ± 39.86	91.47 ± 40.80	0.912
	CI for mean	68.25–111	70.61–115.17	66.17–96.49	75.95–106.99	
Plasma creatinine	Mean	5.14 ± 2.40	5.03 ± 2.9	4.02 ± 1.84	4.4 *	0.523
	CI for mean	4.04–6.23	3.70–6.35	3.32–4.72	2.88–5.22	
Hemoglobin	Mean	10.71 ± 2.28	10.73 ± 2.24	11.56 ± 1.79	11.59 ± 2.00	0.169
	CI for mean	9.68–11.75	9.64–11.81	10.86–12.25	10.83–12.35	
WBC—total count	Mean	8.33 ± 2.15	8.29 ± 0.44	8.55 ± 2.21	8.73 ± 0.74	0.546
	CI for mean	7.35–9.31	7.37–9.22	7.7–9.4	7.69–9.78	
Platelet count	Mean	229.57 ± 72.06	247 ± 81	238.21 ± 68.14	232 *	0.866
	CI for mean	196.77–262.37	208.26–286.37	211.79–264.64	208–247.17	
Potassium *	Median	4.5	4.60	4.8	4.70	1.000
	CI for median	4–4.75	4.27–4.95	4.3–5	4.19–4.90	
Uric acid	Mean	6.87 ± 1.46	6.61 ± 2.05	6.96 ± 1.65	6.65 ± 2.17	0.939
	CI for mean	6.21–7.53	5.67–7.54	6.33–7.58	5.82–7.47	
Phosphorus	Mean	4.44 ± 1.43	4.32 ± 0.98	4.29 ± 1.02	4.25 ± 0.92	0.811
	CI for mean	3.79–5.09	3.88–4.77	3.9–4.67	3.91–4.61	
Albumin	Mean	3.71 ± 0.48	3.72 ± 0.49	3.8 ± 0.64	3.59 ± 0.62	0.433
	CI for mean	3.5–3.9	3.49–3.94	3.58–4.06	3.35–3.82	
hsCRP *	Median	3.66	2.47	2.11	3.32	0.535
	CI for median	1.86–5.67	0.79–5.33	1.25–4.51	1.55–6.06	
LDL-cholesterol	Mean	98.5 ± 45.44	94.1 ± 47.7	80 *	76.54 ± 24.79	0.134
	CI for mean	77.89–119.25	72.43–115.86	62.83–108.5	66.92–86.14	
Triglycerides	Mean	169.48 ± 74.66	152.6 ± 72	204.28 ± 107.73	144 *	0.687
	CI for mean	135.49–203.98	119.84–185.40	163.3–245.25	111.67–207.17	
Sodium	Mean	134.90 ± 3.99	135.10 ± 3.13	135.24 ± 3.52	135 *	0.722
	CI for mean	133.03–136.77	133.03–136.77	133.9–136.58	133.45–136.55	
eGFR (W—CGF) *	Median	23.24/16 + 19.01	24.57/17 + 19.97	28.93/19 + 25.39	25.65/20 + 19.54	0.723
	CI for median	12.35–21.33	12.67–26.57	13.67–33.17	12.67–26.33	
eGFR (A—MDRD) *	Median	15.9/12 + 12.87	17.38/12 + 14.85	22.90/15 + 22.31	19.52/14 + 14.86	0.491
	CI for median	7.67–14.98	9.02–19.61	10.83–21.66	10–23.17	

* Nonparametric variable (Mann–Whitney test). * Median/confidence interval − median. Note: CGF, Cockcroft–Gault formula; MDRD, Modification of Diet in Renal Disease.

**Table 3 nutrients-14-03804-t003:** Interquartile ranges of PCS and IS for Placebo and Enzobiotic groups.

Toxin	Placebo	Enzobiotics
PCS Day 0	10.773–24.135 μg/mL	14.837–31.822 μg/mL
PCS Day 90	13.844–27.959 μg/mL	9.592–19.035 μg/mL
IS Day 0	6258–32,081 ng/mlL	6672–24,498 ng/mL
IS Day 90	6001–23,925 ng/mL	6960–20,799 ng/mL

Lower ranges of toxins were recorded on Day 90 in the Enzobiotic group.

**Table 4 nutrients-14-03804-t004:** Comparison of toxins between placebo and Enzobiotic groups.

Toxins		Placebo			Enzobiotics		
		Mean	SD (±)	CI	Mean	SD (±)	CI
PCS	Day 90/Day 0	1.27 (*n* = 21)	0.93	0.85 to 1.69	0.77 (*n* = 24)	0.48	0.60 to 0.96
	Day 0 absolute (μg/mL)	18.66	10.99	13.70 to 23.62	22.72	10.34	18.35 to 27.09
	Day 90 absolute (μg/mL)	20.97	13.4	14.87 to 27.07	15.69	9.51	11.68 to 19.70
	Change between Day 90 and Day 0	(+)12%			(−)31%		
IS	Day 90/Day 0	1.2 (*n* = 20)	0.71		1.00 (*n* = 27)	0.5	
	Day 0 absolute (ng/mL)	11,462 *	20,679	7603 to 28,355	11,668 *	13,221	8070 to 23,272
	Day 90 absolute (ng/mL)	12,466 *	34,481	7869 to 18,673	10,888 *	12,804	7339 to 16,847
	Change between Day 90 and Day 0	(+)8.8%			(−)6.7%		

* Median. The eGFR ratio of Day 90 to Day 0 for the placebo group showed a reduction of 7% (ratio = 0.93) for IS, whereas it was maintained at the same level (ratio 1.0) in the case of the enzobiotic. In the case of PCS, the ratio of Day 90 to Day 0 for the enzobiotic group was significantly different (*p* = 0.012). The change in the Day90/Day0 ratios of PCS and IS for the placebo and enzobiotic groups is given in Appendix A: Appendix A.

**Table 5 nutrients-14-03804-t005:** Proportion of subjects with relatively bad quality of life scores (adversity ratio).

QOL Components	Number of Questions	Adversity Ratio		Standard Deviation	
		Day 0	Day 90	Day-0	Day-90
Enzobiotic Group					
Daily activity limitations	10	0.2284	0.1003	0.020	0.0150
General wellbeing	2	0.4884	0.1875	0.0539	0.0436
Health	4	0.4012	0.2278	0.0373	0.0334
Emotional problems in last 4 weeks	6	0.4380	0.2333	0.0309	0.0273
Feelings in last 4 weeks	10	0.4047	0.2481	0.0237	0.0216
Problems in last 4 weeks	4	0.4750	0.2597	0.0395	0.0353
Overall	36	0.3726	0.2000	0.0123	0.0105
Placebo Group					
Daily activity limitations	10	0.2382	0.0320	0.0239	0.0111
General wellbeing	2	0.6563	0.2200	0.0594	0.086
Health	4	0.4375	0.2400	0.0438	0.0427
Emotional problems in last 4 weeks	6	0.4896	0.1800	0.0361	0.0314
Feelings in last 4 weeks	10	0.4375	0.2600	0.0277	0.0277
Problems in last 4 weeks	4	0.6667	0.3052	0.0436	0.0472
Overall	36	0.4263	0.1832	0.0146	0.0129

The proportion of adverse scores was lower in the enzobiotic group for four out of the seven components. Accordingly, the QoL as assessed by the adversity index was significantly reduced (*p* = 0.00) in the enzobiotic group compared to the placebo group.

**Table 6 nutrients-14-03804-t006:** The correlation of PCS and IS with uremic parameters.

No.	Parameters	Regression Equation PCS Day 0	*R^2^*	*p*-Value
1	HR	5.44 + 0.4977 HR − 0.003919 HR^2^	4.12%	0.190
2	PR	7.01 + 0.4476 PR − 0.003544 PR^2^	4.01%	0.197
3	PC	6.49 + 0.06809 PC − 0.000049 PC^2^	7.75%	0.044 *
4	RBC	10.22 + 2.32 RBC + 0.030 RBC^2^	3.33%	0.186
5	BUN	17.97 + 0.1803 BUN − 0.002787 BUN^2^	2.66%	0.385
6	Creatinine	16.97 + 2.456 Creat − 0.3384 Creat^2^	5.61%	0.249
7	Urea	17.96 + 0.0843 Urea − 0.000607 Urea^2^	2.67%	0.385
8	UA	22.47 − 3.998 UA + 0.4949 UA^2^	15.31%	0.005 *
9	HDL	35.49 − 0.5509 HDL + 0.003407 HDL^2^	4.44%	0.133
10	eGFR (W-CGF)	13.19 + 0.4523 EGFR(W) − 0.004626 EGFR(W)^2^	7.53%	0.623
11	P	4.26 + 6.128 P − 0.5534 P^2^	1.81%	0.614
12	hsCRP	18.59 + 0.0860 hsCRP + 0.00531 hsCRP^2^	6.59%	0.066
13	Albumin	−2.17 + 16.29 Albumin − 2.717 Albumin^2^	4.07%	0.330
		Regression equation IS Day 0		
1	HR	−3025 + 476 HR − 2.827 HR^2^	0.22%	0.944
2	PR	−3830 + 494 PR − 2.911 PR^2^	0.28%	0.930
3	PC	31,509 − 115.7 PC + 0.2056 PC^2^	1.29%	0.717
4	RBC	58,127 − 15,756 RBC + 1256 RBC^2^	9.99%	0.068
5	BUN	−1191 + 550.6 BUN − 2.123 BUN^2^	18.94%	0.004 *
6	Creatinine	756 + 1143 Creat + 436.0 Creat^2^	50.65%	0.000 *
7	Urea	−1182 + 256.3 Urea − 0.460 Urea^2^	18.94%	0.004 *
8	UA	−23,027 + 11,446 UA − 790.3 UA^2^	2.33%	0.542
9	HDL	27,254 − 444 HDL + 4.01 HDL^2^	0.37%	0.907
10	eGFR (W-CGF)	34,676 − 994.4 EGFR(W) + 6.904 EGFR(W)^2^	32.46%	0.000 *
11	P	7922 − 909 P + 599.6 P^2^	15.25%	0.014
12	hsCRP	15,276 + 510.2 hsCRP − 15.65 hsCRP^2^	2.05%	0.583
13	Albumin	−36,814 + 40,140 Albumin − 6733 Albumin^2^	11.87%	0.037 *

* Significant at α = 0.05. The relationship of PCS with PC (platelet count) and UA (uric acid) was significant in predicting the initial PCS (Day 0). The relationship of IS with BUN, creatinine, urea, and eGFR (Cockcroft–Gault formula) was significant in predicting the initial IS (Day 0). The minimum constants of PCS (Day 0) and IS (Day 0) varied for each parameter; therefore, the significance of combined diagnostic variables for predicting PCS (Day 90) and IS (Day 90) was determined using multiple linear regression.

**Table 7 nutrients-14-03804-t007:** Principal components for each variables.

Variable	PC1	PC2	PC3	PC4	Variable	PC1	PC2	PC3	PC4
HT	0.097	−0.112	0.016	0.173	Hg (hemoglobin)	0.261	0.073	−0.258	0.123
WT	0.184	0.212	0.261	−0.013	HMT	0.280	0.095	−0.218	0.116
BMI	0.133	0.268	0.253	−0.131	Albumin	0.212	−0.131	−0.300	0.112
Temp	0.125	0.063	0.001	−0.178	Na	0.158	−0.164	0.050	−0.141
HR	−0.034	−0.065	−0.297	−0.357	K	0.115	0.100	0.092	0.243
PR	−0.032	−0.075	−0.300	−0.355	Urea	−0.279	0.194	−0.109	0.149
RR	0.038	−0.082	0.056	−0.031	UA	−0.036	0.096	0.008	0.191
Systolic	0.026	0.215	−0.135	−0.022	TP	0.165	0.092	−0.192	0.107
Diastolic	0.102	0.064	−0.226	−0.106	Triglycerides	0.189	0.207	−0.161	0.048
PC	−0.014	0.030	0.146	−0.158	TC	−0.005	0.416	−0.111	−0.189
WBC	0.134	0.156	−0.008	−0.096	EGFR(A- MDRD)	0.278	−0.060	0.132	−0.098
RBC	0.239	0.125	−0.255	0.059	LDL	−0.018	0.403	−0.108	−0.193
SGOT	0.127	0.155	0.053	0.345	HDL	−0.102	0.144	0.110	−0.118
SGPT	0.163	0.066	0.054	0.337	EGFR(W-CGF)	0.289	−0.028	0.174	−0.120
Total bilirubin	0.136	−0.077	0.174	−0.060	HBA1 c	0.039	0.293	0.179	−0.122
BUN	−0.279	0.194	−0.110	0.149	P	−0.232	0.176	0.046	0.146
Creatinine	−0.306	0.104	−0.114	0.065	hsCRP	0.045	0.181	0.249	−0.121

PC1–PC4, principal components. The linear combinations of the factors are shown as PC1–PC4. Variables with the highest negative contrast were taken as predictors.

**Table 8 nutrients-14-03804-t008:** Prediction of PCS (Day 0 and Day 90).

Term	Coef	SE Coef	95% CI	*t*-Value	*p*-Value	VIF
Prediction of PCS (Day 0) as a function of platelet count, uric acid, and creatinine						
Constant	−5.97	7.73	(−21.48, 9.55)	−0.77	0.443	
PC	0.0453	0.0185	(0.0081, 0.0825)	2.45	0.018 *	1
UA	2.987	0.873	(1.233, 4.741)	3.42	0.001 *	1.03
Creat	−1.31	0.642	(−2.600, −0.020)	−2.04	0.047 *	1.04
Prediction of PCS (Day 90) as a function of platelet count (Day 0) and PCS (Day 0)						
Constant	22.92	6.46	(9.77, 36.06)	3.55	0.001 *	
PC (Day 0)	−0.0542	0.0239	(−0.1029, −0.0056)	−2.27	0.03 *	1.02
PCS (Day 0)	0.283	0.128	(0.021, 0.544)	2.2	0.035 *	1.01
Placebo	5.7	3.17	(−0.74, 12.15)	1.8	0.081	1.01

* Significant at α = 0.05.

**Table 9 nutrients-14-03804-t009:** Significance of creatinine prediction from uric acid and platelet count according to age based on multiple linear regression.

	*p-*Value	Coefficient	SE Coefficient
Age	0.0000	−0.04082	−0.0081
Uric Acid	0.0000	0.2904	0.0498
Platelet Count	0.0180	−0.000003	−0.000001

**Table 10 nutrients-14-03804-t010:** IS (Day 90) prediction after treatment (enzobiotics and placebo) as a function of initial creatinine, urea, and phosphorus values.

Term	Coef	SE Coef	95% CI	*t*-Value	*p*-Value	VIF
Constant	−19,551	7873	(−35,607, −3494)	−2.48	0.019	
Creatinine (Day 0)	4689	2320	(−42, 94.20)	2.02	0.052	4.83
Urea (Day 0)	−242.7	91.5	(−4292, −56.2)	−2.65	0.012	3.17
Phosphorus (Day 0)	3946	2313	(−771, 8664)	1.71	0.098	1.85
IS (Day 0)	1.206	0.227	(0.744, 1.668)	5.32	0.000	1.83
Randomization Placebo	3281	5083	(−7086, 13,647)	0.65	0.523	1.23

## Data Availability

Data are archived and can be produced on demand.

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
