# Peer review of "Enzobiotics—A Novel Therapy for the Elimination of Uremic Toxins in Patients with CKD (EETOX Study): A Multicenter Double-Blind Randomized Controlled Trial"

_nutrients, 2022, doi:10.3390/nu14183804_

Round 1

Reviewer 1 Report

This is a very interesting and important study.  The Conclusions are supported by the study design and Results.  Please define all abbreviated terms at the beginning of the manuscript.  In the schema, larger print is needed.  I was not able to understand the statistical difference, if any, on why the plasma creatinine was higher at Day 90 compared to Day 29.  Make this more clear in the text under the schema.  The authors nicely pointed out the strengths and limitations of the study.  I would accept his manuscript.

Author Response

Referee 1  Comments and Suggestions for Authors

This is a very interesting and important study.  The Conclusions are supported by the study design and Results.

Reply:   Thank you for your comment

 Please define all abbreviated terms at the beginning of the manuscript. 

In the schema, larger print is needed. 

I was not able to understand the statistical difference, if any, on why the plasma creatinine was higher at Day 90 compared to Day 29.  Make this more clear in the text under the schema.

Reply:  Thanks  There is a little bit of misunderstanding here. Creatinine did not increase at day 90 it is actually  the number of patients  that is  29.

 The authors nicely pointed out the strengths and limitations of the study. 

I would accept his manuscript.

Reply: Thank you for your comment

I would accept his manuscript.

Reply:  Thank you for your comment

Reviewer 2 Report

Even though the idea sounds really interesting, there are some important points that need clarification, refinement, reanalysis, rewriting, and more information to improve this article. The abstract, introduction, drawing, description of results, and discussion need to be improved to achieve this purpose.

 Major points

1.      The manuscript needs writing and editing, and the title needs a little change, “Enzobiotics, A Novel Therapy for the Elimination of Uremic Toxins in Patients with CKD (EETOX Study): A Multicenter Double-Blind Randomized Controlled Trial”.

In the abstract: The first time an abbreviation appears (highlighted in light blue), the full name must be entered, such as Chronic Kidney Disease (CKD). Do you mean estimated glomerular filtration rate: eGFR? What was the main aim of this study? The main objective must be direct and the same throughout the manuscript (abstract, introduction, results/discussion). Write only one decimal for numbers and percentages. It would be better not to show the standard deviations here. Authors should not use the words that appear in the title as keywords. References should be recent, relevant, and referenced correctly.

2.      In the introduction section:  It would be good if the authors show how they achieved the objective of this randomized clinical trial, taking into account the importance of the microbiome and toxin production in CKD patients (with and without hemodialysis) to establish the main objective. It is not clear what the authors know about this specific topic, it would be better to develop a little more on this axis (microbiome/toxins/inflammation). It would be good to delve into the importance of these factors in CKD. (Nallu, A., Sharma, S., Ramezani, A., Muralidharan, J., & Raj, D. (2017). Gut microbiome in chronic kidney disease: challenges and opportunities. Translational research: the journal of laboratory and clinical medicine179, 24–37). How does this situation happen? Does it depend on the diet, the microbiome, or the stage of the disease? Under what circumstances do these toxins predict the worsening of CKD? Are these PBUTs used as biomarkers today? Earlier studies have targeted the dysbiotic gut, to improve uremic toxins in CKD. How? Line 56: Which authors, in which population, when was it carried out, and under what conditions? Line 60-62: So this would be the next step. Line 79: it would be better to use this information in the discussion section. It would a good idea to show the title after the Figures. Figure 1 should clearly show the words (improve quality). In the first part of Figure 1, the authors describe what "generally" happens with PCS and IS in a patient with CKD (at all stages?), right? In the second part of this figure, the authors show what "could happen" with the "Enzobiotic" treatment (hypothesis/introduction), or show what happened according to the results obtained (results/discussion). Therefore, Figure 1 should be modified accordingly. The research question should be clearly outlined. A good and clear justification for conducting this study should be given. It would be better if the authors offered a clear hypothesis before the main objective of this study. Why would this study be crucial? What was the first endpoint, the secondary endpoint?

3.      The materials and methods section needs deep improvements. More details are required to be able to replicate this study. The description of the study design should be clear, concise, and detailed. It would be a good idea to use more subsections, such as Population Study, Ethical Considerations, Sampling, Clinical (Anthropometric) Evaluation, etc. In Figure 2, what happened to the cardiac parameters? It is part of the 34 diagnostic variables studied. Figure 2 (CONSORT flow diagram) should improve resolution. In this figure, the authors said that this multicenter study covers diversified Indian population settings to reduce bias and increase reproducibility. How did they do that? The authors must declare that this study follows the Declaration of Helsinki (ethical considerations). Line 85: which centers, which states? What does log mean here? What treatment effects were considered? Avoid repetition of data. How was this double-blind study developed? The population under study (characteristics) must be defined. Clearly define all outcomes, exposures, predictors, potential confounders, and effect modifiers. It must be clear who obtained the information and by what means (questionnaires), from where the information was obtained (medical history, face-to-face), whether or not there was a clinical evaluation of the participants, at what time of the year the information was obtained, and if the population was classified by gender (How many women participated?), by BMI, by age groups, by having diabetes or not, etc. The description and quality of the figures of the methods (it is hard to read) to obtain PCS and IS in the supplementary Table 1 should be improved.  What were the cut-off points used to assess these 34 variables studied? All variables should be defined and measured appropriately. What does SRI Mumbai mean? Line 147: is "16" a reference? Line 148: What do these 36 questions assess? Was its analysis quantitative or qualitative? Line 150: What components? What cut-off points for PCS and IS were used? Please explain in the manuscript a little more about the results described in the supplementary material. In statistical analysis, all tests should be described, such as the nonparametric Mann-Whitney median test with 95% CI. How did the authors control for bias?

4.      In the results section: What were the most significant results? Table 2 is confusing, why did the authors decide to show the results in mean/median/SD/CI? This clinical parameter (Lines 184-185) should be described in the material and methods section. Table 3 had no title. t would be a good idea to show in this table the percentage increase/decrease in PCS and IS after 90 days of treatment. Lines 192-215: It would be better to show these results in a table to be able to see the difference between the two groups studied. Lines 217-219: These sentences should be written in predictive analytics in the M&M section. Figures 3 and 4 should be improved. Lines 229-231: this was defined in the M&M section. In the abstract authors said “QOL, as assessed by adversity ratio, reduced significantly (P=0.00) in Enzobiotic group compared to Placebo”  but in the results section the authors reported, “Both the groups had the worse scores at baseline and both groups improved at Day-90”. Lines 239-241: This description should be written in the M&M section. Lines 242-244: This is part of the discussion section. In Table 4, avoid repeating these values if they appear in the table. Where is Supplementary 2?

5.      The discussion should be more argumentative. This section should start with the primary objective of this study and the most significant result. What are the protein metabolism defect and dysbiosis in these patients? Line 323: what are the authors trying to convey? Lines 331-332: this phrase is repeated. Lines 334-337: Please summarize these data better. Lines 342-345: this idea is repeated before. It would be better to write it before so that they can explain about the Enzobiotic (it should not be written in bold). It would be a good idea not to show the "-" or "+" symbols because the authors already explain that there was "a reduction" or "an increase" in the values studied. Line 349: I would like to know if this improvement (positive change = reduction) was statistically significant for IS in the same way as for PCS. If the authors specified CKD patients with a specific diet during this trial, this recommendation and the type of diet should be described in detail in the intervention subsection in the M&M section. In the discussion section, the authors should explain how this diet plan contributes with the Enzobiotic to the improvement of the condition of these patients, would it be part of the strategy to follow? Did all patients follow this diet? Line 364: was it the same for the control group or not? Is there any relationship between PCS/IC with abnormal creatinine values and with other markers in the worsening of patients with CKD? Line 384: this result should be shown in the M&M section. Line 386: it would be better “could translate”. Line 389-391: what were the urea and creatinine levels in these patients? PCS and IS levels are markers that can be used instead of urea and creatinine to predict the worsening of CKD. To what extent do they do it? The results must be discussed from multiple angles and placed in context without being over-interpreted.  How does this prediction equation help in the clinic? In the limitations of this study, what happened to the bias? A paragraph of suggestions should be written before the conclusion. 

6.      The conclusion needs to improve. Review the conclusion based on the results of this study in this specific population. This paragraph could be written before the limitations of this study. In the discussion section, the authors should explain better and more in-depth (Figure 1?) how this therapeutic option could delay the progression of the disease, improve the quality of life and improve the management of anemia in CKD.

Minor points are highlighted in the accompanying manuscript.

 I encourage the authors to rewrite the manuscript, thinking about the principal goal of this study, and its design, and answering with the results and arguments of the discussion the most proper conclusion to this research work.

Author Response

Referee 2  Comments and Suggestions for Authors

Even though the idea sounds really interesting, there are some important points that need clarification, refinement, reanalysis, rewriting, and more information to improve this article. The abstract, introduction, drawing, description of results, and discussion need to be improved to achieve this purpose.

 Major points

  1. The manuscript needs writing and editing, and the title needs a little change, “Enzobiotics, A Novel Therapy for the Elimination of Uremic Toxins in Patients with CKD (EETOX Study): A Multicenter Double-Blind Randomized Controlled Trial”.

Reply : Thank you for your comment : Accepted We have done it

Referee In the abstract: The first time an abbreviation appears (highlighted in light blue), the full name must be entered, such as Chronic Kidney Disease (CKD).

Reply : Thank you for your comment. Done it as chronic kidney disease (CKD)

Referee Do you mean estimated glomerular filtration rate: eGFR?

Reply : Yes eGFR

Referee What was the main aim of this study? The main objective must be direct and the same throughout the manuscript (abstract, introduction, results/discussion).

Reply : The primary objectives were to evaluate efficacy and safety of Enzobiotics in reducing generation of PCS and IS, stabilizing renal function, improving quality of life (QoL) and secondary objective was to evaluate  feasibility of diagnostic prediction of IS and PCS from CKD parameters.

Referee Write only one decimal for numbers and percentages. It would be better not to show the standard deviations here.

Reply: Done it Tables are attached as separate file also.

Authors should not use the words that appear in the title as keywords. References should be recent, relevant, and referenced correctly.

Reply: pcresol, indoxyl sulfate, eGFR; Protein-bound uremic toxins (PBUT’s); Adversity- Ratio; SF-36-QoL

      References are in place. 

Referee In the introduction section:  It would be good if the authors show how they achieved the objective of this randomized clinical trial, taking into account the importance of the microbiome and toxin production in CKD patients (with and without hemodialysis) to establish the main objective. It is not clear what the authors know about this specific topic, it would be better to develop a little more on this axis (microbiome/toxins/inflammation). It would be good to delve into the importance of these factors in CKD. (Nallu, A., Sharma, S., Ramezani, A., Muralidharan, J., & Raj, D. (2017). Gut microbiome in chronic kidney disease: challenges and opportunities. Translational research: the journal of laboratory and clinical medicine179, 24–37). How does this situation happen? Does it depend on the diet, the microbiome, or the stage of the disease? Under what circumstances do these toxins predict the worsening of CKD? Are these PBUTs used as biomarkers today? Earlier studies have targeted the dysbiotic gut, to improve uremic toxins in CKD. How? Line 56: Which authors, in which population, when was it carried out, and under what conditions? Line 60-62: So this would be the next step. Line 79: it would be better to use this information in the discussion section.

Reply Introduction is  Revised Due to word limit the introduction was cut short.

  1. Armani RG, Ramezani A, Yasir A, Sharama S, Canziani MEF, Raj DS.Gut Microbiome in Chronic Kidney Disease. CurrHypertens Rep. 2017 Apr;19(4):29.gastro.2006.12.054 this paper was reference number 6 in the earlier submitted The language was changed to avoid plagiarism. However I have inserted excerpts from this publication now in the

Revised Introduction:

Despite advances in understanding mechanisms responsible for causing renal disease and advent of newer therapies for controlling modifiable risk factors, decline in renal function is still inevitable. Traditional risk factors cannot completely explain renal outcomes in  

Chronic Kidney Disease (CKD) patients. Intestinal microbiota has emerged as an important cause for progression and complications of chronic kidney disease (CKD)[1]. Chronic kidney disease is a chronic uremic state, a perfect prescription for toxin formation and development of cardiovascular disease contributing significantly to total mortality in dialysis patients. Of the many stimulants to prevent progression of CKD, and cardio-vascular disease (CVD), dysbiotic gut microbiome has emerged as a prominent cause of generation of uremic toxins which precipitate progression of cardiovascular disease.[2]

Uremic toxins are biomarker associated with decreased renal function. p-Cresol sulfhate (PCS) and Indoxyl sulfate (IS) contribute to CKD progression through renal tubular damage or tubulointerstitial fibrosis by activating free radical production, upregulating nuclear factor (NF)-κB and plasminogen activator inhibitor type 1, and enhancing the expression of transforming growth factor beta 1, tissue inhibitor of metalloproteinase, and pro-alpha 1 collagen [3]

In addition, IS is reported to be related to aortic calcification, vascular stiffness, along with an increased risk of overall and cardiovascular (CV) mortality in patients with CKD through mechanisms of increasing oxidative stress in endothelial cells, shedding of endothelial microparticles, impairing endothelial cell repair, and inducing vascular smooth muscle cell proliferation. [4]

The first study to support renal toxicity of p-cresol sulfhate (PCS) indicated that p-cresol sulfhate (PCS) was capable of resulting in renal tubular cell damage by inducing oxidative stress by activation of NADPH oxidase [5] a similar mechanism caused by Indoxyl sulfate IS [6,7]  In pre-end stage renal disease (ESRD) patients, their results were further supported by those of Chi-Feng etal [8], that PCS level can predict cardiovascular event as well as kidney function deterioration. These studies explicitly indicate that PCS is not only a vascular toxin but also a nephrotoxin.

Prolonged retention of undigested protein in the intestines triggers immune response and increased inflammation in the gut and formation of uraemic toxins like p-cresol sulfate (PCS) and Indoxyl sulfate (IS), which play an important role in the genesis of cardiovascular complications, progression of renal damage and mortality in chronic kidney disease. Of all the toxins known so far, Indoxyl sulfate (IS) and p-cresol sulfhate (PCS) have been reported to be involved in the development of cardiovascular disease.

 p-Cresyl sulphate (PCS) and indoxyl sulphate (IS) are prototypic protein-bound uraemic toxin molecules, which are not only biomarkers for renal function but have been shown to contribute to the development of diseases [2][ The two toxins are  very similar in their origin from the gut bacteria and their working as  promoters  of renal disease. Both are toxic products of protein metabolism [9], and are bound to albumin at Sudlow II site [10] with low dialytic clearance.  Both the toxins play significant role in renal metabolism, [1,11] cardiovascular disease and mortality in renal patients [12,13]

The classical sources of uremic solutes such as dietary protein breakdown, alternative sources such as environment, herbal medicines, if not restricted create uremic toxicity. Many solutes with toxic capacity are produced in the intestine [14] (most research on uraemic toxicity has focused on retention and removal of these organic compounds[15]

 During natural progression of CKD, there is a shift in microbiome composition and the intestinal environment from a symbiotic to a dysbiotic state caused by an increase in colonic protein fermentation and resulting in increase in microbiota-derived uremic toxins along with diminution of carbohydrate fermentation and consequently formation of shorty chain fatty acids (SCFAs) is compromised [16,17,18]

The amounts of nutrients entering the colon mainly depend on dietary intake and the efficiency of the assimilation process in the small intestine. Pancreatic inflammation and malabsorption are often associated with impaired protein digestion in ESRD population [18] and consequently, changes in the composition of intestinal flora, or changes in intestinal production and absorption, alters their serum concentration. Undigested proteins and dysbiosis of gut in CKD generate IS from tryptophan and PCS from tyrosine [19,20] These protein bound uremic toxins (PBUTs) contribute to worsening renal function and CKD progression [21]. PBUTs also increase CVD morbidity and mortality in CKD [21].

Earlier studies by Feng, Barretto [13]and I Wen-Wu etal [22] had shown PBUTs (PCS and IS) role as a nephrotoxin, vasculotoxin causing worsening aortic calcification, pulse wave velocity contributing to progression of CKD, cardiovascular morbidity and mortality. I-Wen-wu et’al established that PCS and IS may predict the risk of renal progression. The progression of CKD stages increases with increase in toxins [22], Understanding dysbiotic microbiome accompanied by reduced renal clearance, accumulating undigested protein, enhances generation of toxins in blood, resulting in end-stage-renal-disease (ESRD). Although this report enumerates descriptive analysis between stages evidencing PCS 10mg/L as progressive cases  at eGFR of 26 but with PCS 4 mg/ml as non progressive cases at eGFR of 48, but  the predictive equation developed as the facility to test toxin has not been available in many settings.

Enzobiotics modulate gut microbiota and enhance absorption of proteins in the small intestine to potentially prevent formation of protein bound uremic toxins (PBUT’s) from Intestinal microbial metabolism of aromatic amino acids, and promote muscle building and improve muscle recovery.

Since inflammation and oxidative stress are evident in the moderate stages of CKD, the key hypothesis that controlling toxin levels can reduce CKD complications and slow CKD progression, therefore, clinical trials should be planned to see if early intervention, can prevent formation of uremic toxins and benefit quality of life of CKD patients.

Why would this study be crucial?

The gut microbiome plays important roles in both the maintenance of health and the pathogenesis of disease. Gut microbiome dysbiosis, results from alteration of composition and function of the gut microbiome and disruption of gut barrier function, which is commonly seen in patients with chronic kidney disease (CKD). The dysbiotic gut microbiome generates excessive amounts of uremic toxins, and the impaired intestinal barrier permits translocation of these toxins into the systemic circulation and cause progression of CKD and increased cardiovascular risk. Therapeutic interventions should aim to restore gut microbiome symbiosis. If proven effective, these interventions will have a significant impact on the management of CKD patients [21]

Undigested Proteins from small intestine get fermented in dysbiotic colon triggering immune response inducing inflammation. PCS from Tyrosine, and IS from Tryptophan induces inflammation contributing to CKD complications Figure 1. Earlier studies have targeted dysbiotic gut, to improve uremic toxins in CKD. However, targeting impaired protein assimilation (digestion and absorption) in CKD [23] and unmetabolized protein moves to colon to generate PBUTs [24] and never been addressed sufficiently. Results of the studies by the same group on Synbiotics (probiotic + prebiotic) significantly improved serum urea, creatinine, hsCRP, TNF-alfa and quality of life in stage in dialysis patients [25]. Another study established the role of Proteolytic enzymes in metabolizing proteins and improving protein assimilation which was reflected as higher serum albumin levels in peritoneal dialysis patients [26]. A preclinical trial  established superiority of Enzobiotics (Synbiotics+Proteolytic enzymes) over synbiotics alone  and proteolytic enzymes alone in gentamycin induced CKD in Wister rats[27]. Hence, the rationale that Synbiotics and Proteolytic enzyme as one (Enzobiotics) would synergize the action of the both [27]. The favourable results of preclinical trial motivated the authors  to conduct a double-blind, randomized controlled clinical trial on CKD patients to evaluate role of Enzobiotics in reducing PBUTs. In CKD (PCS, IS) and improving QOL, The primary objectives were to evaluate efficacy and safety of Enzobiotics in reducing generation of p-cresol sulfate (PCS) and indoxyl sulfate (IS), stabilizing renal function, improving quality of life (QoL) and secondary objective was to evaluate  feasibility of diagnostic prediction of IS and PCS from CKD parameters.

Referee It would a good idea to show the title after the Figures. Figure 1 should clearly show the words (improve quality). In the first part of Figure 1, the authors describe what "generally" happens with PCS and IS in a patient with CKD (at all stages?), right? In the second part of this figure, the authors show what "could happen" with the "Enzobiotic" treatment (hypothesis/introduction), or show what happened according to the results obtained (results/discussion). Therefore, Figure 1 should be modified accordingly. 

Reply: Done it please read the legend and the paragraph from introduction given below.

Figure 1.Dysbioticgut with inadequate protein metabolism contributes to increased inflammatory state responsible for CKD progression, morbidity and mortality. Supplementation of Enzobiotic (Synbiotics+Proteolytic Enzymes) with food ensures complete protein metabolism in aneubiotic environment, thus, reducing inflammatory state and  better outcome in  CKD rat model

In the second part of this figure, the authors have shown how enzobiotic works (mode of action)which is the basis of the work. Hence we have modified the legend as “Dysbioticgut with inadequate protein metabolism contributes to increased inflammatory state responsible for CKD progression, morbidity and mortality. Supplementation of Enzobiotic with food ensures complete protein metabolism in aneubiotic environment, thus, reducing inflammatory state and  better outcome in  CKD rat model.

To clarify further paragraph in introduction “The studies of the authors on Synbiotics (probiotic + prebiotic)significantly improved serum urea, creatinine, hsCRP, TNF-alfa and QOL in CKD [13]. Proteolytic enzymes metabolize proteins and improve protein assimilation reflected as higher serum albumin levels [14].A preclinical trial showed superiority of enzobiotics (Synbiotics+Proteolytic enzymes) oversynbiotics and proteolytic Enzymes in ratmodel. Hence, the rationale that Synbiotics and Proteolytic enzyme as one (Enzobiotics) would synergises the action of the both [15].The favourable results of Preclinical trial were the next step on human trial presented in this paper in reducing PBUTs.”

Referee The research question should be clearly outlined. A good and clear justification for conducting this study should be given. It would be better if the authors offered a clear hypothesis before the main objective of this study. Why would this study be crucial? What was the first endpoint, the secondary endpoint?

REPLY Changes have been done in the manuscript.

 Why would this study be crucial? The gut microbiome plays important roles in both the maintenance of health and the pathogenesis of disease. Gut microbiome dysbiosis, results from alteration of composition and function of the gut microbiome and disruption of gut barrier function, which is commonly seen in patients with chronic kidney disease (CKD). The dysbiotic gut microbiome generates excessive amounts of uremic toxins, and the impaired intestinal barrier permits translocation of these toxins into the systemic circulation and cause progression of CKD and increased cardiovascular risk. Therapeutic interventions should aim to restore gut microbiome symbiosis. If proven effective, these interventions will have a significant impact on the management of CKD patients

  1. ArmaniA. RamezaniA. YasirS. SharamaM. CanzianiD. Raj Gut Microbiome in Chronic Kidney Disease Current Hypertension Reports Medicine, Biolog y2017

Undigested Proteins from small intestine get fermented in dysbiotic colon triggering immune response inducing inflammation. PCS from Tyrosine, and IS from Tryptophan inducesinflammation contributing to CKD complications Figure 1. Earlier studies  have targeted dysbiotic gut, to improve uremic toxins in CKD. However, targeting impaired protein assimilation (digestion and absorption) in CKD [11] and unmetabolized protein moves to colon to generate PBUTs [12] and never been addressed sufficiently. The studies conducted by the authors on Synbiotics (probiotic + prebiotic)significantly improved serum urea, creatinine, hsCRP, TNF-alfa and QOL in CKD [13]. Proteolytic enzymes metabolize proteins and improve protein assimilation reflected as higher serum albumin levels [14]. A preclinical trial showed superiority of enzobiotics (Synbiotics+Proteolytic enzymes) over synbiotics and proteolytic Enzymes in rat model. Hence, the rationale that Synbiotics and Proteolytic enzyme as one (Enzobiotics) would synergises the action of the both [15].The favourable results of Preclinical trial motivated the authors  to conduct a double-blind, randomized controlled clinical trial on CKD patients to evaluate role of Enzobiotics in CKD reducing  reducing PBUTs. (PCS, IS) and improving QOL.

 for this human trial presented in the paper for 

The primary objectives were to evaluate efficacy and safety of Enzobiotics in reducing generation of p-cresol sulfate (PCS) and indoxyl sulfate (IS), stabilizing renal function, improving quality of life (QoL) and secondary objective was to evaluate  feasibility of diagnostic prediction of IS and PCS from CKD parameters.

Referee The materials and methods section needs deep improvements. More details are required to be able to replicate this study. The description of the study design should be clear, concise, and detailed. It would be a good idea to use more subsections, such as Population Study, Ethical Considerations, Sampling, Clinical (Anthropometric) Evaluation, etc.

Referee In Figure 2, what happened to the cardiac parameters? It is part of the 34 diagnostic variables studied.

Reply: cardiac parameters were included in 34 variables and Table 2 distolic blood pressure. Refer to file cardiac parameters.

 Figure 2 (CONSORT flow diagram) should improve resolution. In this figure, the authors said that this multicenter study covers diversified Indian population settings to reduce bias and increase reproducibility. How did they do that? The authors must declare that this study follows the Declaration of Helsinki (ethical considerations). Line 85: which centers, which states? What does log mean here? What treatment effects were considered? Avoid repetition of data. How was this double-blind study developed? The population under study (characteristics) must be defined. Clearly define all outcomes, exposures, predictors, potential confounders, and effect modifiers. It must be clear who obtained the information and by what means (questionnaires), from where the information was obtained (medical history, face-to-face), whether or not there was a clinical evaluation of the participants, at what time of the year the information was obtained, and if the population was classified by gender (How many women participated?), by BMI, by age groups, by having diabetes or not, etc.

Reply:  All the points raised have been included in the manuscript. Patient Evaluation: The medical history was taken face-to-face in the out-patient-department.

Physical examination was done by the consultant. Vitals were taken  by nurse. Information was captured on case report forms. Patients were enrolled between March 2019 and August 2019.

 The description and quality of the figures of the methods (it is hard to read) to obtain PCS and IS in the supplementary Table 1 should be improved.  

What were the cut-off points used to assess these 34 variables studied? All variables should be defined and measured appropriately. What does SRI Mumbai mean? Line 147: is "16" a reference? Line 148: What do these 36 questions assess? Was its analysis quantitative or qualitative? Line 150: What components? What cut-off points for PCS and IS were used?

Reply: Added in the manuscript

Biochemical parameters collected from all the centers were tested in a single   NABL accredited laboratory “The SRL - Ranbaxy laboratory” Mumbai- India to avoid bias. Reference ranges (cut offs)  for biochemical parameters of SRL - Ranbaxy laboratory were used.

The cutoff user Risk of progression of CKD and CVD risk in the study was determined at p-cresol > 20µg/ml and for indoxylsulphate a value > 20000 ng/ml. These values are determined as per the multiple regression for prediction of toxin at day 90. Risk of progression of CKD in the study was defined as value of PCS> 20µg/ml for IS> 20000 ng/ml, determined from multiple regression for prediction of toxin at Day-90

To determine the effect of Enzobiotics on Quality of Life, SF36 questionnaire KDoQI was used16. Adversity was defined as a proportion of response relatively bad scores in attending their daily course in QOLfor each of 36 questions. These 36 questions provide the health status on a qualitative scale of scores on Quality of living self-assessed by patients at Day 0 and Day 90 for both groups. 6 components -general,daily activities limitation,recent 4 weeks, problems, recent 4 weeks emotional problems, recent 4 weeks feelings and heath result.thebad scores were analyzed as proportion of adverse scores in each group at Day0 and Day 90.v The analysis verified for internal consistency by cronbachs alpha before comparing the adversity ratio between groups.The standard deviation was also compared for consistency between groups. This was analyzed using Binomial proportion of P- Chart against each question and the reduction due to Enzobiotics was evaluated classifying the questions to 7 components - General,Daily Activities Limitation, Recent 4 weeks problems, Recent 4 weeks emotional problems, Recent 4 weeks feelings, Heath result and overall scores. The bad scores were analysed as proportion of adverse scores in each group at Day0 and Day 90. The analysis verified for internal consistency by cronbach’s alpha before comparing adversity ratio between groups including overall QOL.To authenticate the internal consistency Cronbach alpha was adopted

Please explain in the manuscript a little more about the results described in the supplementary material. In statistical analysis, all tests should be described, such as the nonparametric Mann-Whitney median test with 95% CI. How did the authors control for bias?

Reply The parametric T test for difference of mean was used since the ratio of Day 90/ Day 0 followed normal distribution. The absolute values of p-cresol while followed normality, The IS followed log normal distribution. The risk of proportion of patients beyond cutoff level were estimated accordingly using capability analysis. The non-parametric was used in non-normal cases - of Mann whitney Median test.Ratio of improvement in PCS and IS on Day-90 to Day-0 was taken for Enzobiotics and Placebo. The ratio implies, no change if it is 1, Adverse outcome if >1 and Improvement if <1. The normality and homogeneity of variance (F-ratio) for initial observations (Day-0) between the two groups were carried out for randomness. The cutoff user risk of progression of CKD and CVD risk in the study was determined at p-cresol > 20µg/ml and for indoxylsulphate a value > 20000 ng/ml. These values are determined as per the multiple regression for prediction of toxin at day 90.The improvement in PCS and IS was compared with the normal range using the capability analysis after the therapy for potential risks between the groups. The two-sample T-Test was used to test the significance of difference between the two groups on theDay-90 to Day-0 ratio, after verifying for normality.

The subjects not meeting criteria for analysis were discarded to maintain the veracity and validity of observations for reliable statistical comparison. Reasons are documented in Table 1

Refree In the results section: What were the most significant results? Table 2 is confusing, why did the authors decide to show the results in mean/median/SD/CI? This clinical parameter (Lines 184-185) should be described in the material and methods section. Table 3 had no title. t would be a good idea to show in this table the percentage increase/decrease in PCS and IS after 90 days of treatment. Lines 192-215: It would be better to show these results in a table to be able to see the difference between the two groups studied. Lines 217-219: These sentences should be written in predictive analytics in the M&M section. Figures 3 and 4 should be improved. Lines 229-231: this was defined in the M&M section.

Reply: Corrections have been made tables are redone.

Referee In the abstract authors said “QOL, as assessed by adversity ratio, reduced significantly (P=0.00) in Enzobiotic group compared to Placebo”  but in the results section the authors reported, “Both the groups had the worse scores at baseline and both groups improved at Day-90”. Lines 239-241: This description should be written in the M&M section. Lines 242-244: This is part of the discussion section. In Table 4, avoid repeating these values if they appear in the table. Where is Supplementary 2?

Reply the adversity ratio reduced significantly (P=0.00,) which implies  QOL improved  significantly in ENzobiotic group. I have improved expression in the abstract. Thanks

Referee The discussion should be more argumentative. This section should start with the primary objective of this study and the most significant result. What are the protein metabolism defect and dysbiosis in these patients? Line 323: what are the authors trying to convey? Lines 331-332: this phrase is repeated. Lines 334-337: Please summarize these data better. Lines 342-345: this idea is repeated before. It would be better to write it before so that they can explain about the Enzobiotic (it should not be written in bold). It would be a good idea not to show the "-" or "+" symbols because the authors already explain that there was "a reduction" or "an increase" in the values studied. Line 349: I would like to know if this improvement (positive change = reduction) was statistically significant for IS in the same way as for PCS.

REFEREE  If the authors specified CKD patients with a specific diet during this trial, this recommendation and the type of diet should be described in detail in the intervention subsection in the M&M section. In the discussion section, the authors should explain how this diet plan contributes with the Enzobiotic to the improvement of the condition of these patients, would it be part of the strategy to follow? Did all patients follow this diet? Line 364: was it the same for the control group or not? 

Reply The subjects followed CKD diet prescribed by a renal dietician (protein 0.6g/kg/d; energy 30-35 kcalories/d, sodium 2.0 g/d; Potassim <1mEq /kg/d; phosphorus 800-1000 mg/d; Calcium 1500-2000 mg/d with fluid restriction according to renal function) Diet pattern of the subjects was not changed after recruitment in the study in both Enzobiotic and Placebo groups to prevent introduction of any bias and see the effect of Enzobiotic therapy with CKD diet.

REFEREE Is there any relationship between PCS/IC with abnormal creatinine values and with other markers in the worsening of patients with CKD? Line 384: this result should be shown in the M&M section. Line 386: it would be better “could translate”. Line 389-391: what were the urea and creatinine levels in these patients? PCS and IS levels are markers that can be used instead of urea and creatinine to predict the worsening of CKD. To what extent do they do it? The results must be discussed from multiple angles and placed in context without being over-interpreted.  How does this prediction equation help in the clinic? In the limitations of this study, what happened to the bias? A paragraph of suggestions should be written before the conclusion. 

Reply: WE have explained this in the manuscript. Discussion is detailed  now.

REFEREE The conclusion needs to improve. Review the conclusion based on the results of this study in this specific population. This paragraph could be written before the limitations of this study. In the discussion section, the authors should explain better and more in-depth (Figure 1?) how this therapeutic option could delay the progression of the disease, improve the quality of life and improve the management of anemia in CKD.

REPLY improve the management of anemia in CKD. is taken care of in the discussion

REVISED DISCUSSION:

Chronic kidney disease (CKD), is a multifactorial, progressive debilitating disease. Numerous studies have shown that accumulation of toxic metabolites in blood and other metabolic compartments as a result of their enhanced generation from the dysbiotic microbiome accompanied by reduced renal clearance causes progression of CKD to end-stage renal disease (ESRD). The progression has been shown to be influenced by several other factors, such as dietary intake, mental stress and medications[29,30]

Although medication or renal replacement therapies may delay the progression of CKD[31]other preventable factors such as dysbiosis of gut microbiota are gaining much scientific popularity. Randomized controlled trials on synbiotic therapy, have shown reduction of PCS concentration [32]

The intestinal dysbiosis resulting from the loss of kidney function is associated with the secretion of urea into the gastrointestinal tract and successive hydrolysis of urea by some gut microbes and generation of and accumulation of higher concentrations of ammonia which raises intestinal pH resulting in mucosal irritation and has a negative impact on the growth of commensal bacteria favouring maintenance of intestinal dysbiosis[33]

Dysbiosis of the intestinal microbiota increases urea toxins, such as indole-3 acetic acid, p-cresylsulfate (PCS) and IS[34] via endotoxemia and systemic inflammation, finding their way into the into the blood circulation,  thus inducing microinflammation, renal endothelial dysfunction, fibrosis, and tubular damages, which subsequently accelerates the decline of renal function[22]

In recent years, probiotics have been recognized as an adjuvant therapy for CKD for retarding progression of CKD by regulating the intestinal flora alteration and by reducing formation of uremic toxins by reverting the physiological imbalance of the intestinal microbiota. Few randomized controlled trials (RCTs), have shown that probiotics supplementation may reduce the levels of PCS and elevate the levels of IL-6 thereby protecting the intestinal epithelial barrier of patients with CKD[35] . IS and PCS are considered not as only markers for renal function but also predictors of disease progression[36]

High-serum PCS levels are associated with renal progression and all-cause mortality independent of age, gender, diabetes status (seems to cause a surge in  the concentration of the two metabolites), albumin levels, serum IS, serum creatinine, Ca × P product, intact parathyroid hormone, haemoglobin or high-sensitivity C-reactive protein level[36]  In vitro studies have shown the mechanisms of how high-serum PCS promotes renal progression. PCS significantly increases the percentage of leucocytes displaying oxidative burst activity at baseline[37] PCS also induces a dose-dependent shedding of endothelial micro-particles in the absence of overt endothelial damage[12]  These findings indicate that PCS exerts a proinflammatory effect and can alter endothelial function. The relationship between PCS and cardiovascular disease and mortality has been evaluated in several studies[38,39]

The present study was conducted to examine the response of Enzobiotics supplementation on uremic toxins PCS and IS in pre-dialysis CKD patients. PCS and IS are documented independent markers of renal dysfunction, progression of CKD, cardiovascular morbidity and all-cause mortality [38,40,41]. In a recent meta-analysis of 16 studies on synbiotics with 645 subjects, McFarlane etal [42] and later Megan Rossi et al [43] showed quality of these studies were moderate to poor. The last two decades have struggled to consistently show beneficial effect of pre, pro, synbiotics on PBUTs. Of these 5 investigated prebiotics, 6 probiotics, and 5 synbiotics. Prebiotic, probiotic, and synbiotics supplementation may have led to little or no difference in serum urea (9 studies, 345 participants: mean difference (MD) 20.30mmol/L, 95% confidence interval(CI) 22.20 to 1.61, P=0.76, I2=53%), IS ( 4 studies, 144 participants : MD 20.02mg/dL m 95% CI 20.09 to 0.06, P=0.61, I2=0%) and PCS (4 studies, 144 participants: MD 20.13 mg/dL, 95% CI 20.41 to 0.15, P=0.35, I2=0%). These studies showing no statistically significance benefit of Synbiotics on serum urea, creatinine and Toxin levels. studies looking at microbiota changes showed significantly increased levels of Bifidobacteria with uncertain clinical outcomes. Another meta analysis by Wu etal reported that results of a meta-analysis indicate that elevated levels of PCS and IS are associated with increased mortality in patients with CKD, while PCS is associated with an increased risk of cardiovascular events [44]

Considering the poor quality, inconsistent and insignificant outcomes of earlier trials, this EETOX study is the first of its kind successful attempt, where improved protein assimilation using proteolytic enzyme is used in combination with Synbiotics for restoration of eubiotic state of gut in CKD. The working group of this study have named this combination of proteolytic enzyme with synbiotics as Enzobiotics.

This study had homogeneity of subjects enrolled for Enzobiotics and placebo group (absence of significant difference (p = 0.207) between the placebo (18.7 ±10.9) and Enzobiotic groups (22.7 ±10.3) in the levels of two uremic toxins at baseline) and was consistent. PCS showed significant reduction in the Enzobiotics group. Ratio of Day-90/Day-0, PCS showed significant improvement by 23% reduction with Enzobiotics (showing improvement by 2.35 times) while in the placebo group (n=21) the mean PCS increased by 27 %.   A homogenous improvement was recorded as 1.64 times. Similar trend was seen with IS levels albeit less impressive. The absolute median value reduced by 6.7% in Enzobiotics group compared to an increase by 8.8% in Placebo group. The ratio (Day-90/Day-0) IS in Placebo showed increase of 20% compared to 0.3% among enzobiotic group, indicating benefit of treatment. The difference was striking and homogenous.

Our study shows that Enzobiotics therapy can reduce and or maintain (without causing further increase) the concentrations IS and PCS. For predicting PCS and IS levels in the sera, all the 34 variables were found to be interdependent as they were co-varying (supplementary data 3 Figure no S4 Eigen Loading Plot). Out of 34 variables. 82% of the structural variation between variables was explained by 11 principle variables nine of which were negative contrast and two were positive principle variables. The nine negative contrast variables identified were creatinine. BUN, Urea, phosphorus, HDL, uric acid, heart rate, pulse rate, platelet count, eGFR, and RBC. Two positive variables, the heart rate and the platelet count projected as most significant variables in differentiating between placebo and Enzobiotics groups. Relationship of pCresyl sulfhate  and indoxyl suplfate with 11 predictor variables is shown in supplementary file S-3figure S-4 - Scree Plot and Loading Plot. Concentration of uremic toxins was predicted with these variables with regression analysis. When PC increased by 1 unit and UA by 3 units with creatinine at 1.3 units, p-Cresol increased by 0.05 units. This can be used to predict initial p-cresol. When dealing with any stage of CKD or any other health condition, the gut is of paramount importance Abnormalities in platelet function are central to the development of both thrombotic and haemorrhagic complication [45] and CKD patients are at both the risks a highly increased risk precipitating cardiovascular complications.

A hallmark of CKD is the accumulation of uremic retention solutes in the blood due to decreased kidney function. Which poses potential threat to the cell physiology of blood cells and platelets.  Platelet count as a predictor biomarker predicted 20% reduction from initial PCS level on day 90 of Enzobiotic therapy.  The PCS reduced by 29 µg/ml in placebo and by 23 µg/ml in Enzobiotics

Using multiple linear regression approach EETOX study determined the upper critical limit (UCL) for PCS as 20 µg/ ml and 20000 ng/ml for IS. Enzobiotic group had all patients within the UCL at day-90 irrespective of baseline value unlike Placebo group. No earlier studies had defined these limits.

First time relationship was recorded between PCS- platelet count (p-0.018), uric acid (p0.001) and creatinine (P0.047). These 3 parameters could predict the PCS value. When platelet count was below 2.5 Lac/mm3, the PCS was beyond 20µg/ml at Day-0. At 90 days therapy all subjects in Enzobiotic group had platelet count above 2.5 lac/mm3 and PCS below 20 µg/ml.

The prediction equation was also validated with another data set outside the study group. EETOX study clearly demonstrates direct effects of uremic toxins on platelet counts. Earlier studies have shown inverse relationship of uremic toxins with platelet count [46]. CKD has an increased risk of thrombotic and cardiovascular complications [47,48].

The need for predictive equation for PBUTs was felt necessary to find association with routine kidney laboratory parameters and to use it for prediction, because the test facilities are not freely available in all the hospitals. This was the first attempt .Also, the synbiotics (probitics + prebiotics) with protelytic enzymes synerging together evidenced strong relationship of PBUTS with routine kidney lab parameters establishing changes in PBUTs with change in Uremic acid, Creatine, Platelet count for PCS and change in creatine, urea, phosphorus for IS.

At 90 days 72.58% of the variation in IS could be explained by the creatinine, urea and phosphorus levels. No other study has recorded such an association. RBC count showed negative linear correlation with IS (P=0.022), similar negative linear relationship has been recorded in earlier studies [49,50]. However, with Enzobiotic treatment the slope changes in IS levels was much steeper. These results would translate to better erythropoiesis with Enzobiotics in CKD, however needs validation in future studies.

Enzobiotic use did not change urea and creatinine levels over 90 days period probably suggesting stabilisation of CKD, which may contribute to delay in progression of CKD.

While serum urea was around 90 mg/ dL in both the groups, change in urea changed IS levels. The study showed that with urea below 120 mg/dL IS was below 20000 ng/ mL. Three variables in particular creatinine, PC and RBC had high predictability for potential risk of high uremic toxins levels Studies supporting the predictive value of PCS and IS in patients with CKD [51]. Interpretation of the results may be affected by different cut-off values of PCS and IS, and low albumin concentrations. Systematic review of the literature, and after excluding studies biased by albumin binding considerations concluded that PCS and IS indeed play a role vascular and renal disease progression [52]

The exact mechanisms by which elevated levels of IS and PCS contribute to CVD and mortality, however, have not been elucidated. Studies suggest that IS and PCS may suppress the activity or activated leucocytes, inhibit the release of platelet-activating factor by macrophages, and contribute to endothelial dysfunction and oxidative stress[53-,55

To see the effect of Enzobiotics on hypertension, odds ratio for day 90 showed that hypertension worsened (100%) in the placebo group (OR CI; 0.5 to 8.5) compared to Enzobiotics. Therefore, administration of Enzobiotics reduced serum toxins which subsequently reduces the burden of cardiac function.

Earlier studies [42,43] had also shown better response for PCS than for IS. This clinical trial was done with foresight of reducing toxin levels and improving quality of life in CKD patients. there are no published studies on relationship between PCS and IS and their impact on quality of life using SF36 KDOQI. This study used Cronbach’s Alpha [16] which authenticates the internal consistency (0.9237 for Placebo and 0.9364 for Enzobiotics) and higher reliability. To ensure robusticity and reduce, the patients were advised to adhere to the CKD diet plan to ensure gut microbiome was not altered with introduction of new food groups in the diet [56]. To see the effect of Enzobiotic supplementation on quality of life in the present study, SF-36 (KDoQI) designed to assess health concepts and transition that are relevant across age, disease, and treatment groups in adults was used  which highlighted the importance of measures that capture concepts beyond biochemical levels.. The impact of intervention on patients with CKD, the SF-36 domains particularly the physical functioning scores domain showed good evidence of responsiveness, higher scores indicated better health status (cronbach’s α coefficient of 0.9056;  lower the score, better it is when Enzobiotics therapy was given, and a cronbach’s α of 0.9381 for placebo but there was no significant difference for lower the better P  0.065 over 90 days between placebo and Enzobiotics groups a. Ninety days supplementation with Enzobiotics therapy showed overall improvement in all the 11 questions consisting of general well- being like improvement in  emotional problems, feelings and health. SF-36 (KDoQI) questionnaire has been validated on Indian CKD patients and results show that the tool is sound, has good internal consistency, convergent validity and discriminant validity.[57]

The QOL consistently and significantly improved in the Enzobiotics group. QOL being one of the indicators for initiation of dialysis (combined physical disability and uremic symptoms), an Improved QOL would translate to delay in initiation of dialysis. However, this inference needs validation on a large pool of patients.

Are There Any Interesting Findings: The study reports four major findings .i) The study has shown that RBC increase as the concentration of IS reduces. If IS exceeds 20000 the RBC reduces below 4.

  1. ii) Creatinine level reduced by 0.3 mg/dL for every million increase in platelet count. Iii) in the absence of laboratory facilities for testing uremic toxins, it is possible to predict levels of IS. When Creatinine increases by 1 mg/dl, IS increases by 1143 ng/ml. The QOL improved in the supplemted group.

Why should probiotics be considered in CKD? Several reasons compel us to think in line of probiotic supplementation to CKD. CKD-related alterations of gut microbial are related to dietary restrictions to prevention of hyperkalemia [58]. In dialysis population, excessive ultrafiltration volumes and/or intradialytic hypotension causing transient intestinal ischemia can aggravate dysfunction and permeability of the gut barrier. Permeable gut barrier promotes translocation of bacteria and endotoxin through the intestinal wall, activate innate immunity and trigger a local inflammatory process contributing to perpetuating gut barrier damage increase oxidative stress. Frequent use of antibiotics potentially modifies bacterial community adding to imbalance in gut flora.. Studies have shown that Probiotic treatment results in a potentially 22%-28% reductions of PCS in analysis of antibiotic-free patients[59,60] Hence, due credence should be given to the results of probiotic supplementation if antibiotic therapy is being given to the patient. Although no significant changes of  BUN and hemoglobin have been  found in metanalysis of RCTs after treatment with probiotics, yet meta-analysis provides evidence that the probiotics supplementation can reduce PCS and increase IL-6 of patients with CKD.

Some prescription antibiotics  also alter intestinal microflora, while others, such as phosphorus binders and ion exchange resins may additionally slow intestinal transit. PCS and IS are two uremic toxins out of known 80 toxins, that are not removed by dialysis88 while others have shown that the serum concentrations of IS and PCS in patients with CKD are 54 and 17 times higher, respectively, than in healthy individuals, and because they are bound to albumin only approximately 30% are eliminated by hemodialysis89,90 Patients with CKD are also at risk of developing encephalopathy and cognitive impairment. A positive correlation between fibroblast growth factor 23 (FGF-23) and IS serum levels, which may imply the link between this molecule and metabolic bone disease in uraemic patients is reported. Finally, the relationship between IS and CKD-related anemia has been observed since it diminishes erythropoiesis, hampers the activity of erythropoietin and enhances programmed cell death of red blood cells (eryptosis).Increased inflammatory biomarkers in stage 3–4 CKD patients, such as glutathione peroxidase and interleukin-6 are reported. Vascular inflammation is involved in the pathogenesis of thrombotic complications. Controlling inflammation, dysbiosis and  oxidative stress can prevent occurrence of CVD in CKD.IL-6 acts both, as, pro- and anti-inflammatory cytokines depending upon whether it activates signal transducers and activators of transcription (STAT1) . [61,62].

Dysbiosis and formation of uremic toxins is strongly implicated as the  cause of cardiovascular disease (CVD) in CKD .Increased levels of IS are associated with enhanced oxidative stress in endothelial cells, vascular smooth muscle cell proliferation, vascular stiffness, peripheral vascular disease, aortic calcification and overall and cardiovascular mortality in patients with CKD.  IS exerts cardiac profibrotic effect, favouring myocardiocytes hypertrophy and predisposes to atrial fibrillation . Hence, it is presumed that  intervention with Enzobiotics might reduce the CVD burden by recreating balanced of gut microflora.

Direct effects of uremic toxins on platelet function have been described, CKD patients have an increased risk of thromboembolic complications, including myocardial infarction, stroke, deep vein thrombosis, and pulmonary embolism which affect clinical outcome and survival.  This prothrombotic phenotype is ascribed to. CKD-associated dysbiosis (Indolic compounds induced platelet hyperactivity) giving rise to phenomenon known as thrombolome (uremic toxins that enhance thrombosis by increasing tissue factor expression, platelet hyperactivity, microparticles release, and endothelial dysfunction) which is spontaneously induced. In the presence of a low concentration of collagen and thrombin, platelet adhesion and aggregation are enhanced by IS P-selectin and GPIIb/IIIa expression  and a key regulator for platelet activation , “oxidative stress”. Platelet activity is enhanced by IS induced production of ROS-mediated p38 mitogen-activated protein kinase (p38 MAPK)[63]

Prebiotics are the food of the microbiota. obtainable from of food, like fresh fruits, vegetables, and whole grains. In general, a diet that is full of more fresh foods and more fibre is going to give you the best results. decreased protein intake. Since CKD patients have  restrictions on consumption of fruits and vegetables and because of development of hyperkalemia and are advised refined food items to prevent hyper phosphatemia therefore, to maintain gut health a combination of proteolytic enzyme combined with pre and probotics is a plausible way of preventing dysboisis and maintaining gut health, and thus  preventing ckd progression. Studies have shown that  intake of prebiotic inulin, enriched with oligofructose, could significantly reduce serum concentrations of PCS and IXS [64]

In advanced renal failure, removal of ureic toxins is made possible with different type of extracorporeal treatment, but eliminating uremic toxins in early stages of CKD is challenging.   Intestinal production of toxins can be reduced by influencing dietary habits, improving gastroparsis, or by oral administration of absorbants. Colonic transit time is a target in CKD.  Pharmacological therapies  targeting reducing colonic transit times are not available. . However, several therapies— including prebiotics—also reduce colonic transit times, in addition to other effects. The use of pre-probiotics should be started in the early stages of kidney failure and certainly for periods longer than four months37.

Strength of the study

This is the largest randomized double-blind trial of synbiotics in CKD with 50 subjects completing the study. The protocols were strict and strongly adhered to. Statistical analysis has been extensive and unique. For the first time a more rational approach of combining synergistic benefits of proteolytic enzyme with synbiotics (Enzobiotics) has been attempted with encouraging outcomes. The reliability of the data is imminent as the baseline parameters of subjects were comparable in both the groups. In absence of free availability of testing facility for IS and PCS, this study has found associations with routine clinical parameters, and developed the prediction equation, never been recorded in any earlier studies.

.

Summary/Conclusion

Several reasons compel us to think in line of probiotic supplementation to CKD. CKD-related alterations of gut microbial are related to dietary restrictions to prevention of hyperkalemia. Permeable  gut barrier promotes translocation of bacteria and endotoxin through the intestinal wall, activate innate immunity and trigger a local inflammatory process contributing to perpetuating gut barrier damage increase  oxidative stress. Frequent use of antibiotics  in CKD population potentially modifies bacterial community adding to imbalance in gut flora..  Our study shows that enzobiotics therapy can reduce and or maintain (without causing further increase) the concentrations  IS and PCS which can prevent progression of renal failure and may delay time to dialysis. Studies have shown that Probiotic treatment results in a potentially 22%-28% reductions of PCS in analysis of antibiotic-free patients. Our EETOX study is the first of its kind successful attempt, where improved protein assimilation using proteolytic enzyme is used in combination with Synbiotics for restoration of eubiotic state of gut in CKD.

5.2. Lmitations of the study

Only 50 patients were finally analysed in the study and the duration of the study was short considering CKD being a gradually progressive disease over years. It is proposed for further research by authors through randomized block design where each block will be cohorts of patients (pre-existing DM, HT, DM and HTN and none of them) being observed six months before and after intervention for the outcome.

Minor points are highlighted in the accompanying manuscript.

 I encourage the authors to rewrite the manuscript, thinking about the principal goal of this study, and its design, and answering with the results and arguments of the discussion the most proper conclusion to this research work.

Reply The manuscript is completely revised.

Interquartlie ranges of Enzo biotics and placebo

Table 3 Interquartile Range For Placebo And Enzobiotics

Toxin for Day-0, Day-90                                       Placebo                              Enzobiotics

PCS Day-0                           :                            10.773 - 24.135μg/ml                       14.837 – 31.822μg/ml

PCS Day-90                         :                            13.844 – 27.959μg/ml                      9.592 – 19.035μg/ml

IS Day-0                :                            6258-32081ng/ml                             6672- 24498ng/m

IS Day-90                            :                            6001-23925ng/ml                             6960-20799ng/ml

___________________________________________________________________________

Lower range of Toxins were recorded in Day-90 in Enzobiotics group.

Table No :  4  Comparison of Toxins between Placebo and Enzobiotics

Toxins

Placebo

Enzobiotics

Mean

SD (+)

CI

Mean

SD (+)

CI

PCS

Day 90/day 0

1.27 (n=21)

0.93

0.85 to 1.69

0.77 (n=24)

0.48

0.60 tp 0.96

Day o absolute (μgm/ml)

18.66

10.99

13.70 to 23.62

22.72

10.34

18.35 to 27.09

Day 90 absolute (μgm/ml)

20.97

13.4

14.87 to 27.07

15.69

9.51

11.68 to 19.70

Change between day 90 and day 0

(+)12%

(-)31%

IS

Day 90/day 0

1.2(n=20)

0.71

1.00(n=27)

0.5

Day o absolute (ng/ml)

11462*

20679

7603 to 28355

11668*

13221

8070 to 23272

Day 90 absolute (ng/ml)

12466*

34481

7869 to 18673

10888*

12804

7339 to 16847

Change between day 90 and day 0

(+)8.8%

(-)6.7%

* Median
1. The eGFR ratio of Day 90 to Day 0 for Placebo showed a reduction of 7 %(ratio = 0.93), whereas in the case of the Enzobiotics it was maintained(ratio 1.0) the same level in the case of Enzobiotic group
2.In the case of PCS, the day 90 to day 0 ratio for the enzobiotic group it was noted to be statistically significant (p=0.012)

Round 2

Reviewer 2 Report

Even though the manuscript has improved, there are several points that need to be changed before publication (highlighted in the text). There are questions asked in the first revision that has not yet been answered, for example, supplementary material "2" is not mentioned in the text.

I would like to encourage the authors to rewrite the entire manuscript with the two main goals in mind, trying to make it orderly, coherent, and fluid. The writing needs to improve significantly so that readers can better understand what the authors are trying to convey. The discussion is too long and difficult to read. In this section, the authors should not repeat the information given in the introduction section. There are entire paragraphs at the beginning of this section, and several paragraphs with only one or two references, which could look like plagiarism. The objectives and the main significant results are the first points that should be written when the authors start the discussion. The discussion must be coherent and argumentative. The authors have enough significant information in the results that must be argued (for or against) and that supports the hypothesis and its logical conclusion

PS: I would like to know if the supplementary material has been modified. 

Author Response

Even though the manuscript has improved, there are several points that need to be changed before publication (highlighted in the text). There are questions asked in the first revision that has not yet been answered, for example, supplementary material "2" is not mentioned in the text.

Reply: Thanks fior your valuable comments. The supplementary files are mentioned in the text and at the end of the manuscript.Supplenentary File 1 is mentioned in section  2.4. Analysis methods of PCS, IS and Biochemical Parameters Supplementary file  1supplementary file  "2" is mentioned below Table 4 in different sections 4.1. Supplemenatry file 3 is mentioned in Predicted Equations. Supplementary 4 is mentioned in Validation with external data.

     2.4. Analysis methods of PCS, IS and Biochemical Parameters:

The STARD guideline were followed in the diagnostic accuracy of PCS and IS given in supplementary I Table  by NABL accredited Laboratory, Notrox Research Private Limited Bengaluru.

.  .

Predicted Equations: The Diagnostic Prediction given in vide supplementary 3 revealed, when PC decreases below 250, the PCS increases beyond 20µg/ml at Day-0. In Enzobiotics group, after Day-90, when platelet count was between 200 to 400 the PCS reduced to less than 15 µg/mL but in Placebo group, with platelet count below 200 p cresol was above 20 µg/mL.

Validation with external data of large set of patients (n=585) Supplementary 4 of  platelet count as a predictor variable, demonstrated that though creatinine level increased based on age, gender, and CKD stage, the creatinine level reduced by 0.3 mg/dL for every million increase in platelet count. It was found that both  uric acid and platelet count have significant effect on creatinine ( p<0.001) for uric acid (p<0.018).

Supplementary Materials: The following supporting information can be downloaded at: www.mdpi.com/xxx/s1,

Supplementary 1: Quantitation of Indoxyl Sulfate and p-Cresol in CKD Patients;

Supplementary 2 : Changes in PCS and IS Day-90/Day-0 Figure S-1, Figure S-2 : Distribution of PCS for Placebo Day-0 , Day-90 &Enzobiotics Day-0 , Day90 andFigure S-3 :Distribution of IS between Day-0 and Day-90 comparing Placebo and Enzobiotics group.

Supplementary 3: Diagnostic Prediction: Table 1-: Principle Components for Loading Plot, Figure S-4: Eigen Value Scree Plot and Eigen Vector Loading Plot and Figure S-5: shows P Cresol Day-90 on PC ( Platelet count) and PCS base zero for Placebo group.

Supplementary 4:  T-2 : External data validation in N=585 Creatinine on Platelet Counts and Uric Acid according to age and

I would like to encourage the authors to rewrite the entire manuscript with the two main goals in mind, trying to make it orderly, coherent, and fluid. The writing needs to improve significantly so that readers can better understand what the authors are trying to convey.

Reply: Thanks for your valuable comments. Please check the manuscript now, you may find it appealing.

 The discussion is too long and difficult to read. In this section, the authors should not repeat the information given in the introduction section. There are entire paragraphs at the beginning of this section, and several paragraphs with only one or two references, which could look like plagiarism. The objectives and the main significant results are the first points that should be written when the authors start the discussion. The discussion must be coherent and argumentative. The authors have enough significant information in the results that must be argued (for or against) and that supports the hypothesis and its logical conclusion

Reply: We have cut short the discussion. No repeatitions hopefully.

The corresponding author is very particular about plagiarism. I always write in my own framed sentences. That is why the references were not there but they have been added.
